# Distributionally Robust Alignment for Medical Federated Vision-Language Pre-training Under Data Heterogeneity

**Zitao Shuai, Chenwei Wu, Zhengxu Tang & Liyue Shen** *
**University of Michigan**
`{liyues}@umich.edu`

Reviewed on OpenReview: `https://openreview.net/forum?id=hb3ZGvBja4`

## Abstract

Vision-language pre-training (VLP) has emerged as an effective scheme for multimodal representation learning, but its reliance on large-scale multimodal data poses significant challenges for medical applications. Federated learning (FL) offers a promising solution to scale up the dataset for medical VLP while preserving data privacy. However, we observe that client data heterogeneity in real-world scenarios could cause models to learn biased cross-modal alignment during local model training. This would limit the transferability of the federally learned representation model on downstream tasks. To address this challenge, we propose **Fed**erated **D**istributionally **R**obust **A**lignment (FedDRA), a framework for federated VLP that achieves robust vision-language alignment under heterogeneous conditions. Based on client datasets, we construct a distribution family that includes potential test-time domains, and apply a distributionally robust framework to optimize the pre-trained model's performance across this space, thus bridging the distribution gap between pre-training data and downstream applications. To avoid over-fitting on client-specific information, we use anchor representation from the global model to guide the local training, and adopt a two-stage approach to first adapt deep layers before updating the entire network. Extensive experiments on real-world datasets demonstrate FedDRA's effectiveness in enhancing medical federated VLP under data heterogeneity. Our method also adapts well to various medical pre-training methods. Code is available: Official Implementation.

## 1 Introduction

Vision-language pre-training (VLP) learns transferable multimodal representations by extracting latent semantics from large-scale image-text pairs, where the dataset scale largely impacts the performance of the learned model (Oquab et al., 2023). However, scaling up multimodal pre-training datasets is a non-trivial challenge especially for medical applications, due to privacy concerns and regulations of patient data sharing (Ladbury et al., 2023). Recent work has explored federated learning as a solution to leverage data across multiple medical institutions while preserving privacy (Lu et al., 2023).

However, in real-world scenarios, datasets collected from different institutes are always heterogeneous. For example, hospitals in tropical regions receive a high proportion of pneumonia patients, whereas those in colder climates may see more pneumothorax cases (Mendogni et al., 2020). This data heterogeneity is not only a long-standing problem in conventional federated learning (Ghosh et al., 2019; Huang et al., 2022), but also a practical challenge that impedes the deployment of multimodal learning in federated setting, such as medical vision-language pre-training. Current medical VLP methods often focus on learning a modality-shared latent space, where the multimodal training data pairs are well-aligned. However, such learned cross-modal alignments may not be transferable to data from unseen distributions. As shown in Fig. 1, this

---

*coresponding author

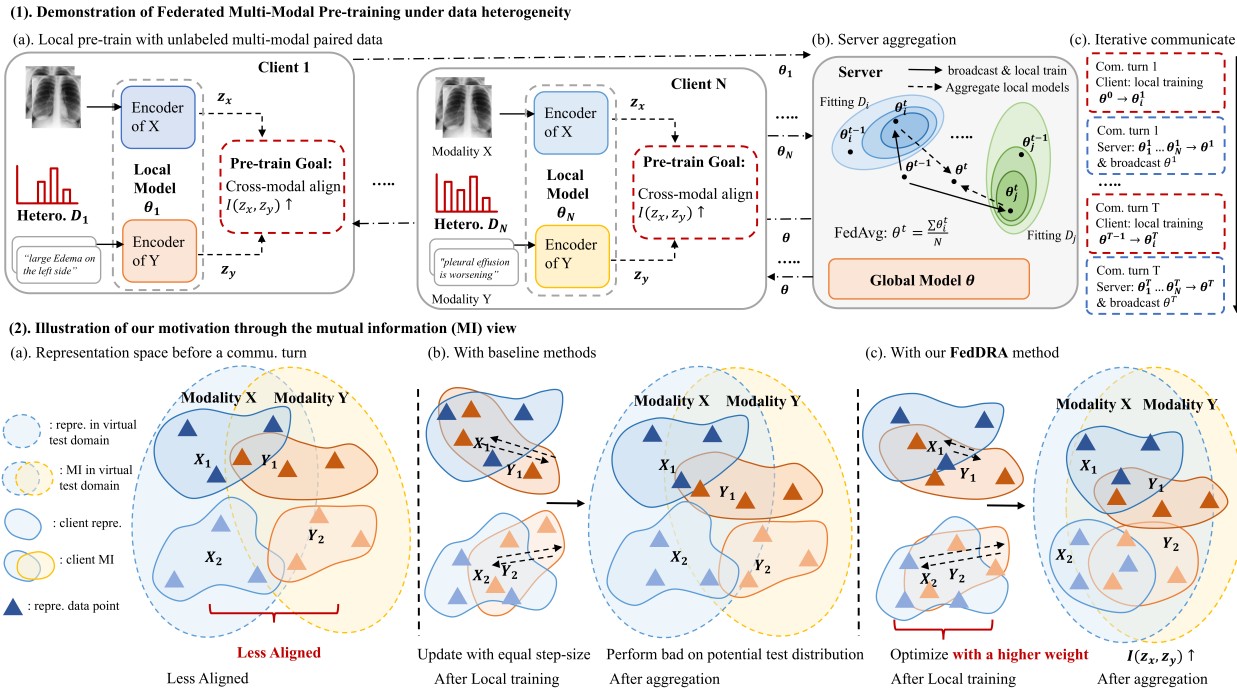

Figure 1: (1) We aim to tackle the data-hungry problem in VLP via federally utilizing heterogeneous multi-modal paired data. During local pre-training, local models pre-trained with naive methods may capture biased client-specific patterns, and simply aggregating them may fail to obtain a robust model. (2). While some clients' data can be easily fitted, cross-modal alignments in other clients are difficult for local models to learn. The simply averaged model would perform worse when the testing distribution is close to less aligned distributions. To balance the fitting of local clients and handle the disparity between training and testing data, our method employs the DRO framework, optimizing model performance over a family of potential testing distributions by dynamically adjusting the update step size of each client.

will harm the performance of the federally pre-trained model, which is aggregated from client models trained on heterogeneous local datasets.

We start by investigating how data heterogeneity affects the performance of federally pre-trained vision-language models. In classic medical VLP (Wang et al., 2022; Bannur et al., 2023), the model learns cross-modal alignment through maximizing the mutual information of the two modalities based on the observed training data. In federated settings, this approach often learns local models that overfit client-specific information. As demonstrated in Fig. 1, simply averaging parameters of biased client models, may obtain a model that performs better on certain over-fitted client data distributions, but fails to robustly generalize to the varying testing domain. Secondly, biased deep layers, which overfit multi-modal correlations of local datasets, would prevent the model from learning transferable and diverse semantics during local training.

In this paper, we propose a **Fed**erated **D**istributionally **R**obust **A**lignment (FedDRA) framework, to learn transferable cross-modal alignment under data heterogeneity. Our key idea is to maximize cross-modal mutual information with distributional robustness. Specifically, to bridge the gap between the downstream testing domain and pre-training data domain, we construct a set of distributions based on client distributions. Then we employ a decentralized distributionally robust optimization method to iteratively improve the pre-trained model's performance on this constructed set. To alleviate the negative effect of over-fitting client-specific information, we maintain a global model to provide anchor representations for guiding local training and utilize a two-stage training scheme to finetune deep layers before updating the whole network.

Our contributions primarily focus on:

- We for the first time tackle the problem of medical VLP under the federated setting by utilizing heterogeneous multi-modal datasets from different institutes. We conduct empirical studies to analyze the influence of data heterogeneity on federated multi-modal learning.

- We propose FedDRA to address the data heterogeneity challenge in federated VLP to obtain transferable cross-modal alignment. It iteratively optimizes model performance on a distribution family and uses a two-stage global-guided local training strategy to reduce over-fitting on client-specific patterns.

- Experiment results show the effectiveness of our method in learning multi-modal representations under the federated setting for various downstream tasks, including image-text retrieval, image classification and segmentation.

## 2 Related Work

**Medical Vision-Language Pre-training.** Pre-training multi-modal models on large-scale datasets and then transferring learned knowledge to downstream tasks has become a popular approach to leverage diverse semantics contained in multi-modal unlabeled data (Li et al., 2022b; Bao et al., 2022; Radford et al., 2021). Current works aim to learn a shared latent space to connect the representations of each modality, leveraging a wide range of self-supervised learning methods, i.e., contrastive learning (Radford et al., 2021; Chen et al., 2020) and multi-modal fusion (Li et al., 2021a; Chen et al., 2022). Medical multi-modal pre-training tasks are often conducted on vision-based datasets, especially vision-language pre-training. Zhang et al. (2022) first utilizes an image-text contrastive loss to align visual and language representations. Huang et al. (2021) aligns fine-grained cross-modal representations through a word-patch contrastive loss, and has improved the performance on fine-grained vision tasks. Furthermore, recent work (Wang et al., 2022; Bannur et al., 2023) incorporates medical domain knowledge to mitigate the misalignment during pre-training. However, almost all of the current methods still rely on large-scale pre-training datasets, which impede their adaptaion to modalities with limited training samples and deployment in real-world scenarios.

**Heterogeneity in Self-Supervised Federated Learning.** Federated self-supervised learning aims to leverage diverse semantics in local unlabeled datasets in a decentralized and privacy-preserved way. One of the key challenges of federated learning is data heterogeneity (Li & Wang, 2019; Collins et al., 2021; Li et al., 2021b; Tong et al., 2023), which has been long discussed in the uni-modality scenarios. Typically, Zhang et al. (2023); Huang et al. (2022); Li et al. (2022a) employ additional communications on local data representations to increase sample diversity. Such methods fail to protect data privacy, which is a vital concern in medical applications. On the other hand, Zhuang et al. (2021); Li et al. (2021b) utilize server model to constrain the update of local models, and Yan et al. (2023) utilizes the mask-autoencoder to handle heterogeneity. Zhuang et al. (2022); Li et al. (2021b); Han et al. (2022) consider distillation-based methods yet ignore the direct modeling of cross-modal alignment. However, these uni-modal self-supervised learning methods have not accounted for the modality gap (Zhang et al., 2024c) between multi-modal data. While uni-modal self-supervised learning aims to learn robust features (Radford et al., 2021), multi-modal learning needs to align input modalities to maximize the mutual information between their representations (Su et al., 2023). Recent advances (Lu et al., 2023) have verified that federated learning can be utilized to scale up the pre-training dataset. However, this work hasn't considered the harm of data heterogeneity issue (Ghosh et al., 2019), which may result in learning biased local model or over-reliance on spurious correlations (Saab et al., 2022) that are client-specific. To tackle this, distributionally robust optimization (Deng et al., 2020) framework can alleviate these issues by optimizing the group-wise worst-case performance on given objective (Liu et al., 2022), which can be flexibly adapted to various of federated learning tasks (Han et al., 2023; Rehman et al., 2023; Capitani et al., 2024).

## 3 Problem Formulation

**Formulation of Pre-training Dataset and Heterogeneity.** In this paper, we consider the multi-modal datasets with two modalities $X, Y$, e.g., image and text modalities. Following (Su et al., 2023), we assume sample $x$ of modality $X$ and sample $y$ of modality $Y$ are generated from latent semantics through implicit

Table 1: Comparison between related works and our proposed FedDRA for federated vision-language pre-training. We organize the related works based on task settings and technical points:(1) Heterogeneous client datasets. (2) Multi-modal paired datasets. (3) Server-computation-free. (4) No communication on training data.(5) Introduce global constraints. (6) Distributionally robust framework.

| Related Work | Task Settings | | | | Related Technical Points | |
|---|---|---|---|---|---|---|
| | Hetero. | Multi-Modal | Server Comp.-Free | Feat. Commu.-Free | Global Const. | Dist. Robust |
| Zhuang et al. (2021) | ✓ | | ✓ | ✓ | ✓ | |
| Zhuang et al. (2022) | ✓ | | ✓ | ✓ | | ✓ |
| Zhang et al. (2023) | ✓ | | ✓ | | ✓ | |
| Yan et al. (2023) | ✓ | | | ✓ | | |
| Lu et al. (2023) | | ✓ | ✓ | ✓ | | |
| Ours | ✓ | ✓ | ✓ | ✓ | ✓ | ✓ |

mappings that are consistent across all clients. For instance, disease labels of a given X-ray image and radiology-report pair, are latent semantic variables that connect these two modalities. That's because these labels determine the pathology region of the radiology image and corresponding description in the diagnosis report. In federated learning, we consider $N$ clients, each has its own local dataset, collectively forming a group $\mathcal{C}$, which represents the set of clients. We assume that each client has a corresponding data distribution $D_i, i \in [1, \ldots, N]$, and data samples are given as $(x, y) \sim D_i$. In real-world scenarios, the distributions $D_i$ of local datasets vary across clients, introducing data heterogeneity that can negatively impact federated learning performance.

We aim to obtain a generalizable model that performs well on the potential testing domain $D_{\mathcal{T}}$. In real-world scenarios, testing datasets often exhibit domain shifts (Zhang et al., 2024b), with samples drawn from distributions distinct from the pre-training data. For example, a medical multi-modal model might be pre-trained on data from routine clinical practice and then transferred to tasks utilizing datasets collected during the COVID-19 pandemic. The overall data distribution $D^{\mathcal{C}}$ could be formed by client datasets $D_i$ with different contribution weights $\lambda_i, i \in [1, \ldots, N]$ [1], when the entire data is grouped by $\mathcal{C}$. Changing these weights alters the aggregated distribution $D^{\mathcal{C}}$. However, these distributions are limited to the pre-training domains and cannot account for unseen cases that may arise during testing. Therefore, we introduce an uncertainty set $Q^{\mathcal{C}}$, which is a distribution family building on the client data distribution while incorporating potential domain shifts. $Q^{\mathcal{C}}$ consists of distributions that are close to $D^{\mathcal{C}}$, and can be formally written as: $Q^{\mathcal{C}} : \{Q : D_f(Q|D^{\mathcal{C}}) \leq \rho\}$, where $D_f$ is the f-divergence of two distributions. Here, $\rho \in R^+$ is the uncertainty radius, and a larger $\rho$ introduces more unseen distributions.

**Federated Vision-Language Pre-training.** Given $N$ clients and their local datasets, federated learning aims to utilize the client dataset to train a generalizable model in a privacy-preserved way. It iteratively trains local models on the client side and aggregates (e.g., FedAvg strategy simply averages model parameters) them on the server. For each communication turn $r$, each client learns a local model through $E$ update steps, and sends it to the server, and overwrites the local model with the aggregated model sent back from the server. Specifically, federated multi-modal pre-training aims to effectively leverage paired and unlabeled multi-modal data from local clients to learn a generalizable model $\hat{f}$.

Multi-modal pre-training utilizes paired data from multiple modalities to learn model $\hat{f}$ that can well represent data samples. In this paper, we consider the pre-training task on image and text modalities. We focus on a conventional setting in vision-language pre-training, where the model consists of two modality-specific encoders for vision input $X$ and text input $Y$, and well-aligned encoders should project their inputs into a shared representation space $\mathcal{Z}$. For example, a good pre-trained model that can encode an image of a running dog and its text description "a photo of running dog" to a shared representation space $\mathcal{Z}$, which is called *cross-modal alignment* in (Castrejon et al., 2016; Gao et al., 2024). Suppose the quality of the representation space of pre-trained models can be measured by a loss objective $\mathcal{R} : \mathbb{R}^{d \times d} \to \mathbb{R}$ (e.g., mutual information between representations $X$ and $Y$ with dimension $d$), a effective federated multi-modal pre-training method should train a model that has a lower risk $\mathbb{E}_{(x,y)) \sim D_{\mathcal{T}}}[\mathcal{R}(\hat{f}(x, y))]$ on the potential testing domain $D_{\mathcal{T}}$.

---

[1]In federated learning scenarios, varying weights can be reflected in the use of different update step sizes or the sampling of different data batches.

Figure 2: (1) Our proposed FedDRA method optimizes performance on a distribution family $\mathcal{Q}^{\mathcal{C}}$ built on client data distributions. After each round of local training, the server performs an additional step to obtain the weight $\lambda_i$ for each client. The combination of $\lambda_i$ and the grouped client distributions constructs a worst-case distribution from $\mathcal{Q}^{\mathcal{C}}$, where the model is expected to perform the worst. The server then transmits $\lambda$ to each client to adjust the local update step size. (2) During local pre-training, we use frozen copies of the server-aggregated models to obtain global representations $z_x^*$ and $z_y^*$, encouraging local models to learn generalizable patterns. Our method adopts a two-stage pre-training schema, which first trains the deep layers, and then jointly optimizes the entire model in the second stage.

In federated setting, $\hat{f}$ is aggregated from $\hat{f}_i$, which are learned by minimizing $\mathbb{E}_{(x,y))\sim D_i}[\mathcal{R}(\hat{f}_i(x,y))]$ during local training. $\hat{f}_i$ may capture client-specific information that may not be generalizable across client dataset domains and will affect the performance of the aggregated $\hat{f}$ if local datasets are heterogeneous.

Table 1 provides a comparison of the most similar previous works, highlighting the distinctions between their tasks and ours, as well as the technical differences between their approaches and ours.

# 4 Method

## 4.1 Global Constrained Local Training Objective

During local training, the pre-trained model would capture client-specific information that can not generalize to other data domains, as shown in Sec. 5.2. Here, we will provide an in-depth analysis of this phenomenon and propose a global constraint term to alleviate it.

In classical vision-language pre-training setting, the vision-language model $f$ is composed of two encoders, $f_\psi : \mathcal{X} \to \mathcal{Z}$ for image modality $X$, and model $f_\phi : \mathcal{Y} \to \mathcal{Z}$ for text modality $Y$, as shown in Fig. 2. Given an image-text paired data $(x, y)$, these models project the input to representations $z_x = f_\psi(x), z_y = f_\phi(y)$. The goal of vision-language pre-training is to learn a robust cross-modal alignment from paired data, and thus obtain a generalizable representation space, where image representation $z_x$ and text representations $z_y$ are well-aligned. It could be viewed as maximizing the mutual information of representations of the two modalities (Su et al., 2023). Therefore, we can measure the cross-modal alignment degree with a mutual-information-based loss objective $\mathcal{R}$, which is approximated by InfoNCE (Liu et al., 2021; Lu et al., 2024) loss in this paper.

Current multi-modal pre-training methods often encourage the model to maximize mutual information of the training pairs, neglecting the potential data heterogeneity problem in the federated learning scenarios. As the distribution $D_i$ varies across clients, each client dataset corresponds to a distinct optimal model $f_i$, which is induced from the distribution of the client dataset. For local training in client $i$, given that only $(x, y) \sim \mathcal{D}_i$ are available, the locally learned encoders $\hat{f}_i$ tends to move towards $f_i$, and might capture some

harmful client-specific information. In federated pre-training, the model is expected to capture patterns that are transferable across clients and potential testing domains, and client-specific information might harm the model's generalization ability. Therefore, it is crucial to explicitly force the model to learn generalizable knowledge. Previous methods such as FedAvg (McMahan et al., 2017; Lu et al., 2023), which do not account for this distinction, may result in learning biased local models $\hat{f}_i$, and diminish the generalization ability of the aggregated model $\hat{f}$.

Our goal is to federally learn a vision-language model that better aligns image and text modalities. Let $f_\psi : \mathcal{X} \to \mathcal{Z}$ be the encoder for the image modality $\mathcal{X}$, and let $f_\phi : \mathcal{Y} \to \mathcal{Z}$ be the encoder for the text modality $\mathcal{Y}$. Suppose the federally learned encoders are denoted as $\hat{f}_\psi : \mathcal{X} \to \mathcal{Z}$ and $\hat{f}_\phi : \mathcal{Y} \to \mathcal{Z}$, and that for each data domain $D_i$, there exist optimal encoders $f_{\psi_i}$ and $f_{\phi_i}$ and learned encoders $\hat{f}_{\psi_i}$ and $\hat{f}_{\phi_i}$. Formally, let $\mathcal{L}$ denotes the InfoNCE, the generalization error on testing domain $D_\mathcal{T}$ can be written as $\mathcal{R}_T(\hat{f}) = \mathbb{E}_{(x,y)\sim D_\mathcal{T}}[\mathcal{L}(\hat{f}_\psi(x), \hat{f}_\psi(y))]$, which is upper-bounded as stated in proposition 1.

**Proposition 1.** *Let $\{D_i, f_{\psi_i}, f_{\phi_i}\}_{i=1}^N$ and $D_\mathcal{T}, f_{\psi \mathcal{T}}, f_{\phi \mathcal{T}}$ be the distributions and optimal encoders for each client data domains and the testing domain, respectively. Given mixed weights $\{w_i\}_{i=1}^N$, $\sum_{i=1}^N w_i = 1$, $w_i \geq 0$, federally learned model $\hat{f}$, and temperature $\tau$ in InfoNCE loss. The generalization error $\mathcal{R}_T(\hat{f})$ follows:*

$$\mathcal{R}_T(\hat{f}) \leq \mathbb{E}_{(x,y)\sim\mathcal{D}_\mathcal{T}} \alpha_i \cdot w_i \cdot \left[ \epsilon_{\hat{f}_\psi, \hat{f}_{\psi_i}}(x) + \epsilon_{f_{\psi_i}, \hat{f}_{\psi_i}}(x) + \epsilon_{\hat{f}_\phi, \hat{f}_{\phi_i}}(y) + \epsilon_{f_{\phi_i}, \hat{f}_{\phi_i}}(y) + C_i \right].$$

where $C_i, \alpha_i$ are client-specific constants, and $\epsilon_{g,h}(x) = \|g(x) - h(x)\|_2^2$ denotes the distance of embeddings of a data sample $x$ projected by $g(\cdot)$ and $h(\cdot)$.

Here, $\epsilon_{f_{\psi_i}, \hat{f}_{\psi_i}}(x)$ and $\epsilon_{f_{\phi_i}, \hat{f}_{\phi_i}}(x)$ measure the discrepancy between the locally trained models and the optimal models of the local data domain. These discrepancies are minimized during local training, leading the local models to learn client-specific information. Another two terms $\epsilon_{\hat{f}_\psi, \hat{f}_{\psi_i}}(x)$ and $\epsilon_{\hat{f}_\phi, \hat{f}_{\phi_i}}(x)$ capture the discrepancy between the server-aggregated models and the locally trained models. Minimizing these terms can not only help reduce the negative effects of data heterogeneity, but also encourage the local models to learn generalizable patterns.

Motivated by this, we can directly minimize the terms $\epsilon_{\hat{f}_\psi, \hat{f}_{\psi_i}}(x)$ and $\epsilon_{\hat{f}_\phi, \hat{f}_{\phi_i}}(x)$ to encourage the models to learn generalizable features. By projecting the inputs $x$ and $y$ through $\hat{f}_\psi$ and $\hat{f}_\phi$, we obtain the global representations $z_x^*$ and $z_y^*$, respectively. Therefore, the constraint loss can be defined as $L_C = ||z_x - z_x^*||^2 + ||z_y - z_y^*||$. The total loss objective for local training could be:

$$L_{local} = L_{MI} + \mu L_C, \tag{1}$$

where $\mu$ is the hyper-parameter that adjusts the constrained degree, $L_{MI}$ is the pre-training loss term based on infoNCE loss (e.g., image-text contrastive loss).

## 4.2 Two-Stage Alignment For Mitigating the Deeper Layer Bias

Furthermore, as discussed in Sec. 5.2, deep layers that contain biased client-specific information can impede pre-trained model to learn generalizable representations. This observation is similar to the findings in the supervised federated learning domain (Legate et al., 2024), where training from a better initialized last layer, can less capture biased information in client local datasets. Motivated by this, we model the deep layers of the encoding functions of modality $X, Y$ as alignment modules $f_\Psi, f_\Phi$, and aim to obtain generalizable alignment modules. In practice, we add additional blocks as the alignment module for simplicity, instead of dividing each encoder to two separate parts. For a data flow of an input pair $(x, y)$, the image encoder and text encoder $f_\psi, f_\phi$ firstly take $x, y$ to obtain intermidiate features $\tilde{z}_x, \tilde{z}_y$, respectively. Then the corresponding alignment modules $f_\Psi, f_\Phi$ will project $\tilde{z}_x, \tilde{z}_y$ to aligned representation $z_x, z_y$, which are finally used in computing $L_{local}$.

To mitigate the negative impact of the biased alignment module on the generalization ability of the pre-trained model, we first adapt alignment modules $f_\Psi$ and $f_\Phi$. During the adaptation, the encoders $f_\psi, f_\phi$ initialized from weights pre-trained in the natural domain are kept frozen. Then we use the finetuned $f_\Psi$ and $f_\Phi$ to enhance the training of the feature encoders $f_\psi$ and $f_\phi$. To encourage alignment modules learn

to extract general features, in the first stage, we train them with frozen feature encoders using the learning objective Eq.(1). Since these feature encoders are less biased from client-specific information, alignment modules are encouraged to learn less biased mappings from $\tilde{z}_y$ to $z_y$ and $\tilde{z}_x$ to $z_x$. Then in the second stage, we train both the alignment module and feature encoders to enhance their capability of extracting medical features, with the same learning objective as the first stage. The complete pipeline is illustrated in Fig. 2.

---

**Algorithm 1** FedDRA for Federated V-L Pre-training

1: **Input:** Init. encoder param. $\theta^{(0)}$; Init. alignment module param. $\Theta^{(0)}$; Client weight $\{\lambda_i^{(0)}\}_{i=1}^N$; Uncertainty set radius $\rho$; Client datasets $\{D_i\}_{i=1}^N$; Local iterations $E$; Step-size $\gamma$.
2: **for** $r = 0, \ldots, R-1$ **do**
3:      Server broadcasts $\Theta^{(rE)}, \theta^{(rE)}, \theta^{(rE)*}, \{\lambda_i^{(r)}\}_{i=1}^N$
4:      **for** client $i = 1, \ldots, N$ **do**
5:          Utilize Algorithm 2 on $D_i$ to get $\Theta_i^{(r+1)E}$
6:      **end for**
7:      Server computes: $\Theta^{(r+1)E} = \frac{1}{N}\sum_{i=1}^N \Theta_i^{(r+1)E}$
8:      Server broadcasts $\Theta^{(r+1)E}$
9:      **for** client $i = 1, \ldots, N$ **do**
10:         Utilize Algorithm 3 on $D_i$ to get $\theta_i^{(r+1)E}$
11:      **end for**
12:      Server computes: $\theta^{((r+1)E)} = \frac{1}{N}\sum_{i=1}^N \theta_i^{((r+1)E)}$
13:      **for** client $i = 1, \ldots, N$ **do**
14:         Compute loss $v_i^{(r+1)}$
15:         of model $(\theta^{(r+1)E}; \Theta^{(r+1)E})$ on $D_i$
16:      **end for**
17:      **for** client $i = 1, \ldots, N$ **do**
18:         $\lambda_i^{(r+1)} = Proj\left(\lambda_i^{(r)} e^{\gamma v_i^{(r+1)}} / \sum_{i=1}^N \lambda_i^{(r)} e^{\gamma v_i^{(r+1)}}, \rho\right)$
19:      **end for**
20: **end for**
21: **return** $(\Theta^{(RE)}; \theta^{(RE)})$

---

**Algorithm 2** 1st Stage Training

1: **Input:** Learning rate $\eta$; Local iteration steps $E$; Local Param. $\theta^{rE}, \Theta^{rE}$; Global Param. $\theta^{rE*}, \Theta^{rE*}$; Weight $\lambda_i$
2: Set $(\theta_i^{rE}; \Theta_i^{rE}) = (\theta^{rE}; \Theta^{rE})$
3: Set $(\theta_i^{rE*}; \Theta_i^{rE*}) = (\theta^{rE}; \Theta^{rE})$
4: **for** $t = rE, \ldots, (r+1)E - 1$ **do**
5:      Sample $\xi_i^t$ from $D_i$ uniformly
6:      $\Theta_i^{t+1} = \Theta_i^t -$
7:      $\lambda_i^r \eta \nabla_{\Theta_i} L_{local}(\Theta_i^t; \theta_i^t; \Theta^{rE*}; \theta^{rE*}; \xi_i^t)$
8: **end for**

---

**Algorithm 3** 2nd Stage Training

1: **Input:** Learning rate $\eta$; Local iteration steps $E$; Local Param. $\theta^{rE}, \Theta^{rE}$; Global Param. $\theta^{rE*}, \Theta^{rE*}$; Weight $\lambda_i$
2: Set $(\theta_i^{rE}; \Theta_i^{(r+1)E}) = (\theta^{rE}; \Theta^{(r+1)E})$
3: Set $(\theta_i^{rE*}; \Theta_i^{(r+1)E*}) = (\theta^{rE}; \Theta^{(r+1)E})$
4: **for** $t = rE, \ldots, (r+1)E - 1$ **do**
5:      Sample $\xi_i^{(t)}$ from $D_i$ uniformly
6:      $(\theta_i^{t+1}; \Theta_i^{t+1}) = (\theta_i^t; \Theta_i^t) - \lambda_i^r$
7:      $\eta \nabla_{(\theta_i; \Theta_i)} L_{local}(\theta_i^t; \Theta_i^t; \theta^{rE*}; \Theta^{(r+1)E*}; \xi_i^t)$
8: **end for**
9: The client sends $\theta_i^{(r+1)E}, v_c^r$ to the server

---

## 4.3 Learning Robust Cross-Modal Alignment via Distributionally Robust Optimization

In real-world scenarios, $\mathcal{D}_{\mathcal{T}}$ is unknown during pre-training and is typically different from the training data distribution, as demonstrated in Sec. 3. To address this issue, we assume the distribution of the testing data is close to that of the entire training data (Rahimian & Mehrotra, 2019; Levy et al., 2020), and thus constructing a distribution family that covers potential testing distributions. As discussed in Sec. 3, the uncertainty set $\mathcal{Q}^{\mathcal{C}}$ comprises distributions that lie within a small distance of the training distribution, making it a suitable choice. We aim to obtain a model that performs well on the whole set of potential testing distributions. Instead of optimizing the model parameter $\theta$ on a distribution randomly selected from $\mathcal{Q}^{\mathcal{C}}$, we optimize $\theta$ over a worst-case distribution from this set, where the pre-trained model performs the worst in aligning the two modalities drawn from it. Formally, this objective can be written as $\mathcal{R}(\theta) := \sup_{Q \in \mathcal{Q}^c} \{\mathbb{E}_{(x,y)\sim Q} [L_{\text{local}}(\theta, x, y)]\}$.

Inspired by this, we introduce the Distributionally-Robust-Optimization (DRO) to our federated multi-modal pre-training task to optimize such worst-case performance. DRO first constructs a family of potential testing distributions $\mathcal{Q}^{\mathcal{C}}$ as shown in Sec. 3, and optimize the model's performance on the worst-case distribution, where the model performs the poorest among distributions in $\mathcal{Q}^{\mathcal{C}}$. However, during federated learning, the server has no access to the distribution of the entire data. Motivated by (Zhang et al., 2024a), we introduce a de-centralized form of the DRO problem. The optimization object could be written as:

$$\sup_{\lambda \in \Delta_{|\mathcal{C}|-1}} \left\{ R(\theta, \lambda) := \sum_i \lambda_i R_i(\theta) \,\text{s.t.}\, D_f(|\mathcal{C}| \cdot \lambda \| (1,1,\ldots,1)) \leq \rho \right\}, \tag{2}$$

where $R_i(\theta) := \mathbb{E}_{(x,y)\sim D_i}[L_{local}(\theta; (x,y))]$ is the empirical risk on client data $D_i$, $\rho$ is the uncertain radius as mentioned in Sec. 3.

Then, we can optimize Eq. 2 by alternatively optimize the weights $\lambda$ and model parameters $\theta$. Specifically, we optimize the parameter $\theta_i$ of local models on each iteration $t$ by $\theta_i^{(t+1)} = \theta_i^{(t)} - \eta \lambda_i^t \nabla_\theta L_{local}$, where $\eta$ is the learning rate. Following the mirror gradient ascent of weight proposed in (Zhang et al., 2024a), we update $\lambda$ with $\lambda_i^{t+1} = \frac{\lambda_i^t e^{\gamma v_i^t}}{\sum_{i=1}^{|\mathcal{C}|} \lambda_j^t e^{\gamma v_j^t}}, i \in [1, ..., N]$, where $\gamma$ is the step-size hyper-parameter, $v_i$ is the loss calculated on $D_i$. Then we compute $\lambda^{t+1}$ by projecting $\tilde{\lambda}^{t+1}$ into the set $\{\lambda : D_f(|\mathcal{C}|) \cdot \|\lambda\| \|(1, 1, \ldots, 1)\| \leq \rho\}$ to fit the constraints of the uncertainty set. In practice, we update $\lambda$ at the end of each communication turn.

We apply the proposed DRO in both the first stage of training the alignment module and the second stage of training the feature encoder. In both stages, we use the same objective, $L_{local}$, to encourage the model to learn generalizable information and mitigate the impact of data heterogeneity on maximizing the mutual information $I(z_x, z_y)$. The key difference between the two stages is, in the first stage, the optimization target $\theta$ in Eq. 2 corresponds to the parameters of the alignment modules $\Phi$ and $\Psi$, whereas in the second stage, $\theta$ represents the parameters of the feature encoders $\phi, \psi$ and alignment modules $\Phi, \Psi$. The pseudo-code of the whole algorithm can be seen in Algorithm 1.

# 5 Experiment

## 5.1 Experiment Setting

We focus on adapting medical vision-language pre-training methods to heterogeneous federated learning settings. We employ the framework of image-text contrastive learning with two modality-specific encoders, a fundamental design in multi-modal pre-training. We use vision-language pre-training tasks on Chest X-ray datasets and ophthalmology image datasets to evaluate the effectiveness of our FedDRA method.

### 5.1.1 Experiment Set-up of Pre-training on Chest X-Ray datasets

**Pre-training setup.** Following (Wang et al., 2022), we utilize the MIMIC-CXR (Bigolin Lanfredi et al., 2022) dataset for medical vision-language pre-training. Following (Yan et al., 2023), we employ the Latent Dirichlet Allocation (LDA) (Blei et al., 2003) to divide the MIMIC-CXR dataset based on disease labels to construct 5 heterogeneous client datasets. We set the heterogeneity degree in the LDA algorithm to be 1. Each divided dataset consists of train splits and test splits based on the notation of the MIMIC-CXR. We use the train split for pre-training, and test split to evaluate pre-trained model's image-text retrieval performance. Here, we only divide the raw data into 5 subsets, because vision-language pre-training requires a large batchsize and is data-consuming, thus we need to guarantee each client has $10k$ to $50k$ paired data.

For main experiments, we set the number of communication turns to 25, and randomly sample 50 batches of data for local training at each turn. Here, we choose a relatively small number of communication turns compared to classical supervised federated learning. That's because VLP needs large local optimization steps per turn to learn cross-modal alignment.

**Downstream tasks.** Following (Wang et al., 2022), we conduct the following downstream tasks to evaluate the transferability and generalization ability of the pre-trained model. **(1) Few-shot classification.** We test their performance on multiple image classification benchmarks: RSNA Pneumonia Detection (RSNA) (Shih et al., 2019), and Covidx (Wang et al., 2020). We fine-tune our pre-trained model with an additional linear layer on 1%, 10% percentage of the training dataset, and evaluate the classification accuracy. **(2) Medical image segmentation.** We conduct medical image segmentation experiments on the RSNA (Wang et al., 2020) benchmark. We freeze the encoder and fine-tune a U-Net decoder using 1%, 10% of the training data, and then use the Dice score for evaluation. The datasets we have used for the fine-tuning based tasks are out-of-distribution, so that we can evaluate the transferability of the pre-trained model. **(3) Image-text retrieval.** We utilize the test splits of client datasets for evaluation, these datasets are unseen in pre-training, and can be viewed as in-domain samples. We report the top-1 and top-5 recall accuracy in a data batch.

### 5.1.2 Experiment Set-Up of Pre-training on Ophthalmology Datasets

**Pre-training setup.** We conduct vision-language multi-modal pre-training using retinal image datasets from different institutes to simulate a more real-world setting. These retinal datasets are from different institutions of low-income and high-income countries, and are highly heterogeneous real-world scenes. Specifically, we utilize MESSIDOR (Decencière et al., 2014) from France and BRSET (Nakayama et al., 2023) from Brazil as pre-training datasets, and assign them to two clients. These datasets include both images and tabular EHR records indicating Diabetic Retinopathy (DR) status and edema risk. For implementation, we transform these tabular data into text captions.

**Downstream tasks.** We evaluate the transferability of the models on few-shot classification tasks using the MBRSET (Wu et al., 2025; Nakayama et al.) dataset. Unlike the pre-training datasets, MBRSET was collected by portable devices, resulting in a significant distribution shift. We perform few-shot classification tasks on diabetic retinopathy and edema status using this dataset. We fine-tune the model with an additional linear layer on 10%, 20% and 100% of the training data, and report classification accuracies.

### 5.1.3 Backbones and Baselines

We focus on enabling medical multi-modal pre-training methods to be applied to heterogeneous federated learning scenes. We have considered the generalization ability of our method on different backbone VLP methods, and adopted contrastive-learning-based methods: simple language-image contrastive alignment (ConVIRT) (Zhang et al., 2022; Radford et al., 2021), global-local language-image contrastive alignment(GLoRIA) (Huang et al., 2021), and Multi-Granularity Cross-modal Alignment (MGCA) (Wang et al., 2022). All of the loss objectives of these pre-training methods contain a contrastive loss term, which act as the infoNCE loss to maximize the mutual information between two modalities. And we take this loss term for computing the client weights in the DRO part.

For baseline federated learning strategies, we have adapted FedMAE (Yan et al., 2023), FedEMA (Zhuang et al., 2022), FedMOON (Li et al., 2021b), FedX (Han et al., 2022), FedU (Zhuang et al., 2021), FedL-DAWA (Rehman et al., 2023) for comparison. These are self-supervised learning methods in the federated learning domain which also focus on tackling the data heterogeneity. For basic federated learning baselines, we consider simple averaging (FedAvg) (McMahan et al., 2017), decentralized training, and centralized training. For baselines pre-trained in Local strategy, we report the averaged performance of the local models.

For fair comparisons, we re-implemented all baseline methods using the same backbones. To adapt uni-modal self-supervised learning baselines to our setting, we added an image-text contrastive loss, applying the same hyperparameters as in our method for consistency. We use ViT-base (Dosovitskiy et al., 2020) as the vision encoder and Bert-base (Devlin et al., 2018) as the text encoder, with input pre-processing following (Wang et al., 2022). Additionally, we employ an extra transformer block from ViT-base and Bert-base as the alignment module for vision and language, respectively. These modules are initialized from the checkpoint pre-trained on natural data.

## 5.2 Empirical Finding

In this section, we will demonstrate our key empirical findings for federated multi-modal pre-training under heterogeneous client datasets, which actually motivates us to propose our FedDRA. We conduct experiments on the image-text retrieval task, which reflects the ability to maximize the mutual information and learn cross-modal alignment. In the following studies, we mainly compare performances of naive federated (FedAvg) pre-trained model, decentralized pre-trained model, and centralized pre-trained model.

**Federated learning enhances pre-training by leveraging more samples in a privacy-preserved manner, while data heterogeneity can affect the effectiveness of the FedAvg.** Figure 3(a) presents the retrieval accuracies of the models under consideration. Despite the heterogeneity of local datasets, the FedAvg strategy significantly outperforms the decentralized pre-training approach. However, the centralized pre-trained model remains an upper bound, indicating substantial room for improvement.

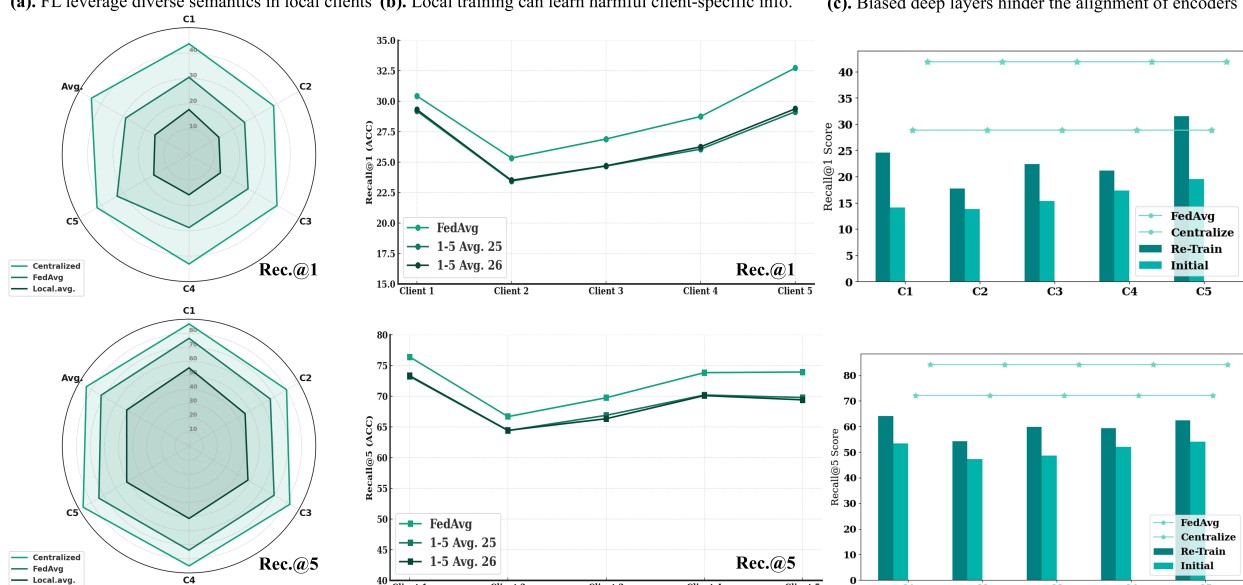

Figure 3: (a). The comparison of retrieval acc. on each client denoted as $\{C_i\}_{i=1}^5$, of centralized, FedAvg, and averaged acc. of decentralized pre-trained models. (b). The performance of the server model after 25 com. turns and the averaged performance of corresponding client models after 25 and 26 com. turns, on each client denoted as $\{\text{Client}_i\}_{i=1}^5$. (c). The averaged acc. on each client. We show the acc. of centralized and FedAvg pre-trained baselines, and de-centralized pre-trained models shown as $\{C_i\}_{i=1}^5$ retrained on the union of training splits of client datasets.

**Local training can learn harmful client-specific information, degrading the performance of the pre-trained model.** After several communication turns, re-training the aggregated server model on local datasets may lead to a performance drop, as shown in Figure 3(b). We considered a set of models retrained on a server model with local datasets respectively. The server model is learned through 25 communication turns. Compared to the starting server model, the averaged accuracy of local retrained models is significantly lower. This degradation may be because local training focus on learning domain-specific information in the late communication rounds, which would affect the aggregated model's overall performance.

**Decentralized pre-trained deep layers can hinder the learning of a generalizable feature extractor.** We re-trained the first four shallow layers of the decentralized pre-trained model on the combined local datasets. While this led to some performance improvements, a significant gap still remains compared to the FedAvg pre-trained baselines, as shown in Figure 3(c). This gap indicates that the biased frozen deep layers prevent the model from learning more diverse semantics from the combined dataset. We hypothesize that these deep layers may contain biased, client-specific information, which obstructs the cross-modal alignment process. Our findings align with observations (Legate et al., 2024) in supervised federated learning.

Overall, from empirical findings, we conclude that federated multimodal pre-training is sensitive to data heterogeneity, and simply averaging local model weights does not solve this issue essentially. Furthermore, performance is closely tied to the generalization ability of the final layers in pre-trained models. Thus, we are motivated to propose our method.

## 5.3 Main Results

**Our method learns robust and enriched cross-modal alignment and has better transferability.** Table 2 has shown results of downstream tasks, here we utilize the ConVIRT as the backbone pre-training method. In the image-text retrievel task, both average and worst-client accuracies of our method are higher than baseline's, which means our model can capture more robust cross-client features. In the few-shot classification and segmentation, our method beats other baseline strategies on each task, which demonstrate the higher generalization ability of the representation space learned by our method.

Table 2: Downstream task performance. We report the few-shot classification accuracy on Covid and RSNA, the in-domain retrieval accuracy, and the Dice score for segmentation on RSNA.

| Strategy | Backbone | RSNA (cls.) | | Covid (cls.) | | RSNA (seg.) | | In-domain Image-Text Retrieval | | | |
|---|---|---|---|---|---|---|---|---|---|---|---|
| | | 1% | 10% | 1% | 10% | 1% | 10% | Rec.@1 | Rec.@5 | Wst.@1 | Wst.@5 |
| FedEMA | ConVIRT | 82.8 | 83.1 | 79.2 | 86.5 | 70.9 | 73.6 | 24.0 | 67.0 | 21.9 | 62.4 |
| FedMOON | ConVIRT | 82.5 | 83.2 | 77.8 | 89.2 | 69.0 | 71.3 | 27.8 | 70.9 | 25.3 | 67.2 |
| FedAvg | ConVIRT | 83.1 | 83.3 | 78.0 | 88.5 | 69.6 | 71.5 | 28.8 | 72.1 | 25.3 | 66.7 |
| FedDRA (Ours) | ConVIRT | **83.2** | **83.7** | **81.0** | **90.3** | **71.7** | **74.1** | **30.2** | **73.2** | **27.0** | **68.9** |
| FedX | GLoRIA | 82.7 | 83.4 | 78.3 | 88.5 | 71.0 | 72.1 | 28.5 | 72.2 | 25.9 | 68.0 |
| FedU | GLoRIA | 83.0 | 83.5 | 78.7 | 89.3 | 71.2 | 72.6 | 29.2 | 73.0 | 27.6 | 69.5 |
| FedAvg | GLoRIA | 83.2 | 83.3 | 77.5 | 89.0 | 71.4 | 72.4 | 29.9 | 73.8 | 27.8 | 69.5 |
| FedDRA (Ours) | GLoRIA | **83.6** | **84.1** | **79.4** | **89.8** | **72.0** | **73.2** | **31.1** | **74.3** | **28.2** | **70.2** |
| FedLDAWA | MGCA | 82.4 | 83.5 | 78.1 | 88.5 | 70.4 | 72.6 | 29.0 | 73.5 | 27.0 | 68.9 |
| FedAvg | MGCA | 82.6 | 83.5 | 75.8 | 88.2 | 70.1 | 71.4 | 29.3 | 73.7 | 26.8 | 70.4 |
| FedDRA (Ours) | MGCA | **83.1** | **83.8** | **79.3** | **89.1** | **71.0** | **72.8** | **29.8** | **74.1** | **27.4** | **70.6** |

Table 3: We have conducted ablation experiments to verify the effectiveness of our key technical designs. We report the few-shot classification accuracy and the retrieval accuracy.

| Two-stage | Global Constraint | DRO-Weighing | Covid (cls.) | | RSNA (cls.) | | In-domain Image-Text Retrieval | | | |
|---|---|---|---|---|---|---|---|---|---|---|
| | | | 1% | 10% | 1% | 10% | Rec.@1 | Rec.@5 | Wst.@1 | Wst.@5 |
| | ✓ | ✓ | 83.0 | 83.4 | 80.5 | 89.6 | 29.4 | 72.7 | 26.2 | 68.1 |
| ✓ | | ✓ | 82.8 | 83.0 | 79.8 | 88.6 | 28.3 | 71.9 | 26.0 | 67.9 |
| ✓ | ✓ | | 82.5 | 82.9 | 80.2 | 89.2 | 29.7 | 72.8 | 25.6 | 67.3 |
| ✓ | ✓ | ✓ | **83.2** | **83.7** | **81.0** | **90.3** | **30.2** | **73.2** | **27.0** | **68.9** |

Table 2 has shown the performance of adapted self-supervised federated learning methods which focus on single-modality. From the results, we observe that baselines have shown better transferability on visual downstream tasks, compared to the naive FedAvg strategy. However, in the multi-modal retrieval task, our method beats these baselines by a large margin, which indicates that previous single-modality methods cannot be easily adapted directly for multi-modal data. Furthermore, FedAvg is still a competitive baseline in multi-modal retrieval tasks compared to other adapted methods, as we observed in the experiments. We conjecture that's because FedAvg only focuses on maximizing the in-domain mutual information, and doesn't introduce additional loss terms which would hurt the learning of enriched cross-modal alignment. However, this may lead to lower generalization ability on few-shot downstream tasks as discussed before.

Table 4: Downstream task performance. We report the few-shot classification accuracy of Diabetic Retinopathy and Edema Risk classification tasks on the MBRSET dataset.

| Strategy | Diabetic Retinopathy (cls.) | | | Risk of Edema (cls.) | | |
|---|---|---|---|---|---|---|
| | 10% | 20% | 100% | 10% | 20% | 100% |
| Decentralized | 78.8 | 79.7 | 81.1 | 91.5 | 92.5 | 93.8 |
| FedAvg | 79.4 | 80.2 | 82.3 | 92.8 | 93.6 | 94.2 |
| FedMAE (Yan et al., 2023) | 79.2 | 80.3 | 82.0 | 92.4 | 93.3 | 94.0 |
| FedX (Han et al., 2022) | 79.5 | 80.1 | 81.6 | 93.0 | 93.5 | 94.3 |
| FedU (Zhuang et al., 2021) | 79.7 | 80.5 | 81.7 | 92.8 | 93.4 | 94.1 |
| FedDRA (Ours) | **80.6** | **81.5** | **83.1** | **93.4** | **94.1** | **94.9** |
| FedGlobal | 81.9 | 82.6 | 84.0 | 94.2 | 94.7 | 95.8 |

**Our method can be transferred to multiple various multi-modal pre-training methods.** Table 2 shows the downstream task performance of the MGCA and GLoRIA backbone pre-training methods when combined with our strategy. Our method has successfully adapted MGCA and GLoRIA to the heterogeneous federated multi-modal pre-training scenario, as demonstrated by the significant improvement in classification and segmentation tasks.

## 5.4 Analysis Experiments

**The two-stage pre-training strategy and global constraints can enhance the learning of cross-modal alignement.** We remove the global constraint loss $L_C$ of our method, and compare the pre-trained model's performances with those of the original version. As shown in Table 3, the downstream performance, particularly the image-text retrieval accuracy, is significantly lower in the modified version. Similarly, we

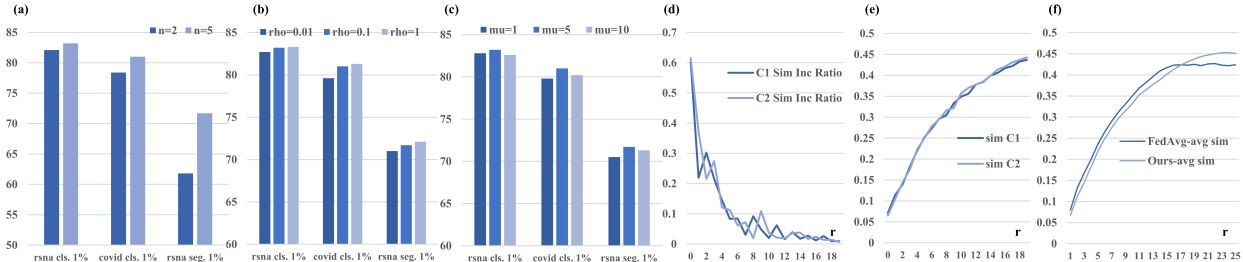

Figure 4: (a) Analysis study on number of clients. (b) Analysis study on uncertainty radius. (c) Analysis study on global constraint degree. (d) Improvement of image-text embedding similarity per.commu turn. (e) Image-text embedding similarity curve of our method. (f) Averaged similarity curves.

remove the first-stage pre-training on alignment modules to verify the role of the two-stage pre-training strategy. We have found that the first stage pre-training can help learn better cross-modal alignment and achieve high image-text retrieval accuracies, as shown in Table 3.

**DRO weighing can reduce the domain gap and improve the downstream performances.** We remove the DRO weighing part and compare the performances of the model pre-trained with the original method. As shown in Table 3, removing the DRO-weighing part leads to a large performance drop in few-shot classification performances, thus adding DRO-weighing component can improve the transferability of the pre-trained model. The client-wise worst accuracies of the original model are much higher than those of the modified version. This means DRO has succesfully bridged the gap between local training data and downstream dataset, by optimizing model performance on the constructed uncertain set of distributions.

Table 5: Results on client datasets with different heterogeneity degrees, across different pre-training methods.

| Strategy | $\alpha$ | Backbone | RSNA (cls.) | | Covid (cls.) | | RSNA (seg.) | | Backbone | RSNA (cls.) | | Covid (cls.) | | RSNA (seg.) | |
|---|---|---|---|---|---|---|---|---|---|---|---|---|---|---|---|
| | | | 1% | 10% | 1% | 10% | 1% | 10% | | 1% | 10% | 1% | 10% | 1% | 10% |
| FedAvg | 1 | ConVIRT | 82.2 | 83.3 | 78.0 | 88.5 | 69.6 | 71.5 | GLoRIA | 83.2 | 83.8 | 78.5 | 89.0 | 71.4 | 72.4 |
| FedDRA (ours) | 1 | ConVIRT | 83.2 | 83.7 | 81.0 | 90.3 | 71.7 | 74.1 | GLoRIA | 81.8 | 82.9 | 78.5 | 89.6 | 69.3 | 72.5 |
| FedAvg | 5 | ConVIRT | 80.7 | 82.3 | 77.6 | 88.2 | 68.5 | 71.7 | GLoRIA | 81.8 | 82.9 | 78.5 | 89.6 | 69.3 | 72.5 |
| FedDRA (ours) | 5 | ConVIRT | 81.8 | 82.9 | 78.5 | 89.6 | 69.3 | 72.5 | GLoRIA | 82.2 | 83.0 | 78.4 | 89.4 | 71.5 | 72.9 |
| Centralized | - | ConVIRT | 83.4 | 84.6 | 82.5 | 92.0 | 72.6 | 76.4 | GLoRIA | 84.0 | 84.7 | 82.2 | 91.8 | 73.5 | 73.7 |

**FedDRA dynamically schedules updating stepsize for each client, and therefore optimizes the worst-case performance.** We select two clients $C_1, C_2$ in the federated pre-training. For each client, we calculate the average cosine similarity between image and text embeddings, using the server-aggregated model at each communication turn. In Fig. 4(d) and Fig. 4(e), we plot curves of these similarities, which reflect the cross-modal alignment degree. At each communication turn, when the similarity of a client is relatively high, its similarity in next turn would get a smaller improvement. That's because our FedDRA can assign higher updating stepsize to client where the cross-modal alignment is less extracted by the model.

**Our FedDRA can alleviate over-fitting client-specific information, and learn better cross-modal alignment** For our FedDRA and the FedAvg method, we plot the cosine-similarities of text and image embedding averaged across clients at each communication turn. As shown in Fig.4(f), FedAvg requires fewer communication rounds to converge, but results in fluctuations after certain communication turns. This aligns with findings in Sec.5.2, where local retraining a model that are trained after multiple communication rounds, would introduce harmful client-specific information and distort the learned representation space. In contrast, our FedDRA gradually extract cross-modal alignment from local training in a distributionally robust manner, and learns a stronger representation space.

**Analysis on global constraint hyper-parameter $\mu$.** As shown in Fig. 4(c), a larger $\mu$ encourages federated pre-training to enhance performance on less optimized client data domains, leading to smaller disparity on image-text retrieval performance on each client domain. As we increase $\mu$ from 1 to 5, the downstream performance consistently increases. However, a excessively large $\mu$ can decrease overall performance.

**A larger uncertainty radius $\rho$ improves transferability in downstream tasks.** Fig. 4(b) shows the downstream performance of models pre-trained with different uncertainty radii in the DRO process. As larger $\rho$ would bring higher performance in few-shot classification and segmentation tasks on out-of-domain datasets. We also observed that a smaller $\rho$ better supports cross-modal alignment learning, achieving better image-text retrieval performance on in-domain datasets, as shown in Table 11 in the Appendix. This is because the larger uncertainty radius would incorporate more potential out-of-distribution cases, which can enhance the model's transferability.

**Robust check on heterogeneity degree of client datasets.** We changed the $\alpha$ which adjusts the heterogeneity degree of the LDA allocated client datasets, to check the robustness of our method under different heterogeneity degree. As shown in Table 5, our method consistently enhance pre-training methods' performances under client datasets with different heterogeneity degrees.

**Robust check on number of clients.** We adjusted the number of clients involved in federated pre-training. As shown in Fig. 4(a), increasing the number of clients introduces greater diversity, which can enhance the downstream performance of the pre-trained model.

## 6 Conclusion

Data limitation is a long-standing problem in the multi-modal learning domain. Despite federated learning can leveraging datasets from multiple sources while guaranteeing privacy issues, its performance would be damaged by data heterogeneity. Inspired by our empirical findings on the impact of heterogeneity on federated multi-modal learning, we propose the FedDRA framework to mitigate heterogeneity for federated medical vision-language pre-training. The effectiveness of our method has been verified by comprehensive experiments. While introducing representation from other clients might bring larger improvement, we still consider the most privacy-preserved setting where representations are not transmissible. Further work could explore how to generalize to more diverse multi-modal pre-training data in federated multi-modal learning, while preserving local data privacy.

## 7 Acknowledgement

The authors acknowledge support from U-M MIDAS PODS Grant and U-M MICDE Catalyst Grant, and computing resource support from NSF ACCESS Program. This work used NCSA Delta GPU through allocation CIS230133 and ELE230011 from the Advanced Cyberinfrastructure Coordination Ecosystem: Services & Support (ACCESS) program, which is supported by National Science Foundation grants 2138259, 2138286, 2138307, 2137603, and 2138296.

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
