# A   Implementation Detailed

## A.1   Details of MIMIC-CXR

### A.1.1   Pre-training setup

Following (Wang et al., 2022) we utilize the MIMIC-CXR (Bigolin Lanfredi et al., 2022) dataset for multi-modal pre-training. This dataset is widely used in the medical multi-modal learning domain, with $227,835$ image-text pairs from $65,379$ patients. Some related works also have imported additional features to image-text pairs to augment the data. However, we only use the image-text pairs for pre-training to make the results and conclusions more generalizable. The MIMIC-CXR dataset is open access, it can be obtained through MIMIC-CXR Access.

During the pre-training, local clients only have access to their highly heterogeneous datasets. To construct the heterogeneous client datasets, following (Yan et al., 2023) we employ the Latent Dirichlet Allocation (LDA) (Blei et al., 2003) to divide the MIMIC-CXR dataset into 5 partitions based on a selected sensitive attribute. For implementation, we import the corresponding attribute information of given image-text pairs from the MIMIC-CXR and divide local datasets based on disease category. The disease category is a multi-label binary attribute and is transformed into a multi-class label. That's because the words in the clinical report are highly related to the disease category as illustrated in Fig 5. We set the heterogeneity degree in the LDA algorithm to be 1 for main experiments. For analysis experiments, we also have run experiments on client datasets allocated by LDA with a heterogeneity degree of 5.

Specifically, we select 5 commonly considered diseases Bannur et al. (2023): 'Edema','Pleural Effusion', 'Consolidation', 'Pneumothorax', and 'Pneumonia'. We set the non-NaN value to 1 and then set NaN value to 0 to construct a 5-way binary multi-label. Then we get $2^5$-category multi-class label and run LDA on them.

**Report Sample 1**:

Previous mild pulmonary edema and possible concurrent pneumonia has all cleared. Heart is top-normal size, improved, and pleural effusions have resolved. Right hilar vessels are still enlarged, perhaps due to pulmonary arterial hypertension. Lateral view shows atherosclerotic coronary calcification in the left circumflex.

**Label:**

**Edema**

**Pneumonia**

**Pleural Effusions**

**Report Sample 2**:

Allowing for differences in technique and projection, there has been little interval change in the appearance of the chest since the previous radiograph, with no new focal areas of consolidation to suggest the presence of pneumonia. Multifocal linear areas of scarring appear unchanged, previously attributed to sarcoidosis. Band-like opacity at periphery of left lung base has slightly worsened and is attributed to localize atelectasis.

**Label:**

**Pneumonia**

Figure 5: Illustration of the strong connection between latent variable and the text modality.

We divide the MIMIC-CXR into 5 heterogeneous subgroups to construct 5 client datasets. Each divided dataset consists of train splits and test splits based on the notation of the MIMIC-CXR. Our pre-trainings are mainly conducted on $4 \times A40$ or $2 \times A100$. The batch size we have utilized ranged from 288 to 388. We set the learning rate to $2 \times 10^{-5}$ in main experiments, the number of communications to 25. For our method, we set the uncertainty radius $\rho = 0.1, \mu = 5$ in main experiments. For each communication, we randomly sample 50 batches of data from the client datasets.

### A.1.2   Downstream tasks

We evaluate the generalization ability of the pre-trained model through three downstream tasks: few-shot classification, medical image segmentation, and image retrieval.

**Few-shot classification.** To assess the model's effectiveness on general medical image tasks, we evaluate it on multiple image classification benchmarks: (1) RSNA Pneumonia Detection (RSNA)Shih et al. (2019), where the task is to predict whether an image shows pneumonia. (2) CovidxWang et al. (2020), which includes three categories: COVID-19, non-COVID pneumonia, and normal. We fine-tune our pre-trained

model with an additional linear layer on 1% and 10% of the training dataset and report classification accuracy on these benchmarks.

**Medical image segmentation.** To explore the model's transferability to fine-grained tasks, we conduct experiments on medical image segmentation using the RSNA Wang et al. (2020) benchmark. Following Wang et al. (2022), we convert RSNA object detection ground truths into segmentation masks. Similar to Huang et al. (2021), we employ a U-Net framework with our pre-trained image encoder as the frozen encoder, while fine-tuning the decoder on 1% and 10% of the training data. The Dice score (%) is used for performance evaluation.

**Image retrieval.** To verify whether the pre-trained models have captured the semantic alignment between image and text in the pre-training data, we perform an image retrieval task. We test image retrieval performance on the validation splits of the local clients. For each text in a batch of image-text pairs, we calculate similarities with images in the batch, then rank these similarities and retrieve the top-1 and top-5 images. If the corresponding image of the text is in the selected set, it is correctly retrieved. We use top-1 and top-5 recall accuracy to evaluate performance.

## A.2 Ophthalmology datasets

### A.2.1 Pre-training setup.

We conduct vision-language multi-modal pre-training using retinal image datasets from different institutes. These retinal datasets are from different institutions of low-income and high-income countries, and are highly heterogeneous real-world scenes. Specifically, we utilize MESSIDOR (Decencière et al., 2014) from France and BRSET (Nakayama et al., 2023) from Brazil as pre-training datasets, and assign them to two clients. These datasets include tabular EHR records indicating Diabetic Retinopathy (DR) status and edema risk. We transform tabular data into text captions in the format: "retinal image with {DR status} and {edema risk}" to obtain text prompts. Similar to MIMIC dataset, our pre-trainings on ophthalmology datasets are mainly conducted on $4 \times A40$ or $2 \times A100$. We set the batch size to 100, the number of communications to 20, and the learning rate to $1 \times 10^{-5}$ in the experiments. For our method, we set the uncertainty radius $\rho = 0.5, \mu = 1$ in main experiments. For each communication, we randomly sample 20 batches of data from the client datasets.

### A.2.2 Downstream tasks.

We evaluate the transferability of the models on few-shot classification tasks using the MBRSET (Nakayama et al.) dataset. Unlike the pre-training datasets, MBRSET was collected in low-income areas using portable devices, resulting in a significant distribution shift. We perform few-shot classification tasks on diabetic retinopathy and edema status prediction tasks using this dataset. These are binary classification problems. We fine-tune the model with an additional linear layer on 10%, 20% and 100% of the training data, and report classification accuracies.

# B Additional Experiment Results

**Federated pre-trained models still show a significant performance gap compared to central-ized pre-trained models in multi-modal retrieval tasks.** Table 6 shows the performance of models pre-trained in decentralized, FedAvg, centralized federated learning strategies, using different backbone pre-training methods. FedAvg has more effectively extract cross-modal alignment from federally utilizing local datasets, and achieved much better transferability on downstream datasets and in-domain image-text re-trieval tasks, compared to de-centralized pre-trained models. However, there are still performance gaps in the retrieval tasks compared to the centralized pre-trained model. That might because each batch of data in centralized pre-training scene has higher diversity, which encourages the contrastive-based model to capture more robust alignment.

Table 6: Downstream task performance on different multi-modal pre-training backbone methods.

| Strategy | Backbone | RSNA (cls.) | | Covid (cls.) | | RSNA (seg.) | | In-domain Image-Text Retrieval | | | |
|---|---|---|---|---|---|---|---|---|---|---|---|
| | | 1% | 10% | 1% | 10% | 1% | 10% | Rec.@1 | Rec.@5 | Wst.@1 | Wst.@5 |
| Decentralized | ConVIRT | 81.5 | 82.3 | 76.5 | 85.6 | 64.6 | 70.7 | 15.5 | 51.1 | 13.6 | 46.0 |
| FedAvg | ConVIRT | 83.1 | 83.3 | 78.0 | 88.5 | 69.6 | 71.5 | 28.8 | 72.1 | 25.3 | 66.7 |
| Centralized | ConVIRT | 83.4 | 84.6 | 82.5 | 92.0 | 72.6 | 76.4 | 41.5 | 84.2 | 38.6 | 80.0 |
| Decentralized | GLoRIA | 82.3 | 82.9 | 77.9 | 86.8 | 71.1 | 72.1 | 17.2 | 52.5 | 15.2 | 48.7 |
| FedAvg | GLoRIA | 83.2 | 83.3 | 77.5 | 89.0 | 71.4 | 72.4 | 29.9 | 73.8 | 27.8 | 69.5 |
| Centralized | GLoRIA | 84.0 | 84.7 | 82.2 | 91.8 | 73.6 | 73.7 | 41.7 | 84.0 | 39.0 | 80.5 |
| Decentralized | MGCA | 81.9 | 82.7 | 77.8 | 87.6 | 62.8 | 70.2 | 15.2 | 50.4 | 13.4 | 45.4 |
| FedAvg | MGCA | 82.6 | 83.5 | 75.8 | 88.2 | 70.1 | 71.4 | 29.3 | 73.7 | 26.8 | 70.4 |
| Centralized | MGCA | 84.0 | 84.5 | 79.5 | 89.5 | 70.7 | 72.5 | 39.9 | 83.5 | 36.9 | 80.3 |

Table 7: Detailed results of downstream task performances.

| Strategy | Backbone | RSNA (cls.) | | Covid (cls.) | | RSNA (seg.) | |
|---|---|---|---|---|---|---|---|
| | | 1% | 10% | 1% | 10% | 1% | 10% |
| FedEMA | ConVIRT | $82.8 \pm 0.32$ | $83.1 \pm 0.17$ | $79.1 \pm 0.12$ | $86.5 \pm 0.27$ | $71.0 \pm 1.55$ | $73.6 \pm 1.08$ |
| FedAvg | ConVIRT | $83.0 \pm 0.49$ | $83.3 \pm 0.36$ | $78.0 \pm 0.43$ | $88.5 \pm 0.51$ | $69.5 \pm 1.72$ | $71.8 \pm 0.84$ |
| FedDRA (Ours) | ConVIRT | $\mathbf{83.2}\pm0.19$ | $\mathbf{83.7}\pm0.12$ | $\mathbf{80.9}\pm0.16$ | $\mathbf{90.3}\pm0.17$ | $\mathbf{71.5}\pm0.95$ | $\mathbf{74.2}\pm0.76$ |
| FedU | GLoRIA | $83.0 \pm 0.36$ | $83.5 \pm 0.22$ | $78.7 \pm 0.44$ | $89.4 \pm 0.26$ | $71.1 \pm 1.28$ | $72.3 \pm 0.67$ |
| FedAvg | GLoRIA | $83.3 \pm 0.45$ | $83.4 \pm 0.23$ | $77.7 \pm 0.40$ | $88.9 \pm 0.48$ | $71.4 \pm 1.43$ | $72.5 \pm 1.05$ |
| FedDRA (Ours) | GLoRIA | $\mathbf{83.6}\pm0.37$ | $\mathbf{84.1}\pm0.19$ | $\mathbf{79.3}\pm0.33$ | $\mathbf{89.8}\pm0.29$ | $\mathbf{71.9}\pm1.31$ | $\mathbf{72.9}\pm0.80$ |
| FedLDAWA | MGCA | $82.3 \pm 0.29$ | $83.5 \pm 0.20$ | $78.1 \pm 0.40$ | $88.4 \pm 0.18$ | $70.0 \pm 1.66$ | $72.2 \pm 1.45$ |
| FedAvg | MGCA | $82.5 \pm 0.48$ | $83.5 \pm 0.28$ | $75.8 \pm 0.67$ | $88.2 \pm 0.35$ | $69.8 \pm 2.13$ | $71.7 \pm 1.47$ |
| FedDRA (Ours) | MGCA | $\mathbf{83.2}\pm0.37$ | $\mathbf{83.8}\pm0.27$ | $\mathbf{79.4}\pm0.12$ | $\mathbf{89.0}\pm0.13$ | $\mathbf{70.8}\pm1.48$ | $\mathbf{72.5}\pm1.15$ |

**Extension to more VLP Methods.** In our paper, we conducted comprehensive experiments on three types of different vision-language pre-training methods including ConVIRT Zhang et al. (2022), GLoRIA Huang et al. (2021), and MGCA Wang et al. (2022), because many previous works have shown the effectiveness of these methods for medical vision-language pretraining by using the self-supervised contrastive learning objective and the dual-encoder network structure. To further validate our method's generalizability beyond contrastive learning, we have incorporated Masked Record Modeling (MRM) Zhou et al. (2023) as an addi-tional backbone with new experiments and results included in the revised manuscript. MRM is fundamentally different from contrastive learning by employing a masked input modeling strategy as self-supervision. As shown in Figure 2 of Zhou et al. (2023), during the pre-training stage, MRM requires the image encoder to provide effective image representations to simultaneously support the restoration of masked image patches and masked associated radiology report tokens.

We adapt FedDRA by applying global constraints on the image encoder, and integrating the DRO framework to dynamically adjust the update step sizes for each client. By using the MRM as the VLP approach, we compared the results of our proposed FedDRA strategy with baselines FedAvg and FedMOON. As demonstrated in Zhou et al. (2023) Appendix A, the MRM pre-training method is designed to learn representations of radiographs specifically for disease diagnosis, it lacks a text decoder, making it unsuitable for image-text retrieval tasks. Therefore, we evaluated the pre-trained model on few-shot classification and few-shot segmentation downstream tasks. The experiment results are summarized in Table 9. As shown in the table, our FedDRA strategy achieves higher performance on all these downstream evaluation tasks with two different datasets. This demonstrates that FedDRA can be extended to a broader scope of pre-training methods beyond contrastive learning-based approaches.

Figure 6: Segmentation downstream task results across multiple runs.

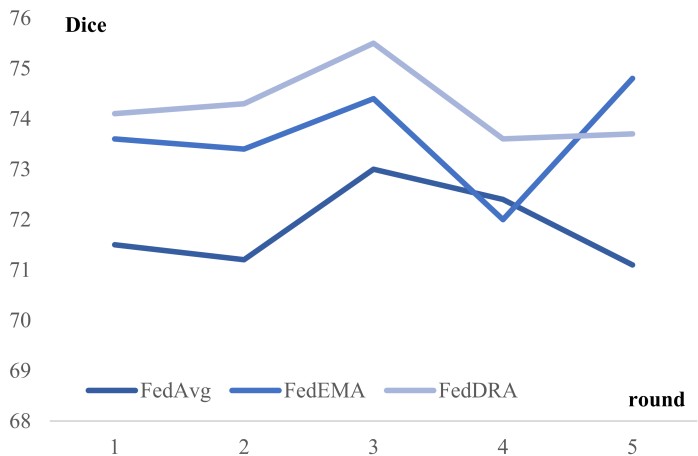

We have provided the computational cost of pre-training. As shown in the following table, we report the number of model parameters and the training time per step, which is averaged over an entire communication round. The experiments were conducted on an Nvidia 4×A40 GPU with a total batch size of 388, using the ConVIRT pre-training backbone. Although our method has a larger number of model parameters, its training time is not significantly higher than that of the baseline methods.

Table 8: Computational cost of different federated pre-training strategies.

| Metric | FedAvg | FedMOON | FedX | FedU | FedEMA | FedLDAWA | FedMAE | Ours [1st + 2nd] |
|---|---|---|---|---|---|---|---|---|
| Averaged Training Time (seconds per batch) | 14.6 | 18.2 | 22.6 | 15.6 | 15.9 | 15.2 | 15.0 | 22.1 |
| Number of Trainable Parameters (M) | 172.6 | 172.6 | 172.6 | 172.6 | 190.4 | 172.6 | 226.1 | 190.4 |
| Additional Module | N | N | N | N | Y | N | Y | Y |
| Using Global Copy During Local Training | N | Y | Y | Y | Y | N | N | Y |

Table 9: Generalization to masked input modeling based pre-training backbone method.

| Strategy | Backbone | RSNA (cls.) 1% | RSNA (cls.) 10% | Covid (cls.) 1% | Covid (cls.) 10% | RSNA (seg.) 1% | RSNA (seg.) 10% |
|---|---|---|---|---|---|---|---|
| FedAvg | MRM | 80.2 | 81.0 | 78.2 | 88.9 | 72.2 | 74.1 |
| FedMOON | MRM | 80.6 | 81.3 | 78.0 | 88.5 | 72.3 | 73.8 |
| Ours | MRM | **81.4** | **82.6** | **78.7** | **89.6** | **72.8** | **74.5** |

## C   Detailed Experiment Results

We have conducted multiple runs of pre-training for both our method and the baseline methods with five total times using different random seeds. We computed the averaged performance and standard deviation based on new results and the original ones. The final averaged performance and standard deviation are presented in Table 7. The results show that our method consistently outperforms the baselines across all tasks after running multiple-time experiments. The averaged performance of our method exceeds that of the best-performing baseline by more than 0.5%, while also exhibiting a lower standard deviation in most of settings, indicating a statistically significant improvement especially for the few-shot classification task on both datasets with three different backbones. For the segmentation tasks, we observe a higher standard

deviation due to the sensitivity of the dice score evaluation metric impacted by the pretrained image encoder, which is also observed for other baseline methods. To better illustrate the significance of our method's performance gain, we have included a plot comparing dice scores of different methods on the segmentation task across multiple runs, as shown in Figure 6. We have compared the 10% few-shot image segmentation results of models pre-trained with FedAvg, FedEMA, and our FedDRA strategy, using the ConVIRT Zhang et al. (2022) backbone.

Here we provide detailed results of ablation studies shown in Fig. 4 in the main text.

Table 10: Ablation studies on the number of clients.

| Num. of Client | RSNA (cls.) | | Covid (cls.) | | RSNA (seg.) | | In-domain Image-Text Retrieval | | | |
|---|---|---|---|---|---|---|---|---|---|---|
| | 1% | 10% | 1% | 10% | 1% | 10% | Rec.@1 | Rec.@5 | Wst.@1 | Wst.@5 |
| n=2 | 82.1 | 83.2 | 78.4 | 88.5 | 61.8 | 71.0 | 23.1 | 62.9 | 19.2 | 57.8 |
| n=5 | **83.2** | **83.7** | **81.0** | **90.3** | **71.7** | **74.1** | **30.2** | **73.2** | **27.0** | **68.9** |

Table 11: Ablation studies on uncertainty radius.

| Uncertainty Radius | RSNA (cls.) | | Covid (cls.) | | RSNA (seg.) | | In-domain Image-Text Retrieval | | | |
|---|---|---|---|---|---|---|---|---|---|---|
| | 1% | 10% | 1% | 10% | 1% | 10% | Rec.@1 | Rec.@5 | Wst.@1 | Wst.@5 |
| $\rho = 0.01$ | 82.7 | 83.2 | 79.6 | 89.1 | 71.0 | 72.8 | **30.4** | **73.5** | 26.6 | 68.4 |
| $\rho = 0.1$ | 83.2 | 83.7 | 81.0 | 90.3 | 71.7 | **74.1** | 30.2 | 73.2 | **27.0** | **68.9** |
| $\rho = 1$ | **83.3** | **84.0** | **81.3** | **90.8** | **72.1** | 74.1 | 28.9 | 72.5 | 26.2 | 67.8 |

Table 12: Ablation studies on global constraint degree.

| Constraint Degree | RSNA (cls.) | | Covid (cls.) | | RSNA (seg.) | | In-domain Image-Text Retrieval | | |
|---|---|---|---|---|---|---|---|---|---|
| | 1% | 10% | 1% | 10% | 1% | 10% | Rec.@1 | Rec.@5 | Disparity |
| $\mu = 1$ | 82.8 | 83.4 | 79.8 | 89.6 | 70.5 | 72.8 | 29.1 | 72.5 | 3.2 |
| $\mu = 5$ | **83.2** | **83.7** | **81.0** | **90.3** | **71.7** | **74.1** | **30.2** | **73.2** | 2.9 |
| $\mu = 10$ | 82.6 | 83.2 | 80.2 | 90.2 | 71.3 | 72.9 | 29.6 | 72.8 | **2.4** |

Here we provided detailed results for our empirical study in Sec. 5.2.

Table 13: The comparison of retrieval acc. on each client denoted as $\{C_i\}_{i=1}^5$, of centralized, FedAvg, and averaged acc. of decentralized pre-trained models using the ConVIRT backbone. We report the local models of decentralized pre-training strategy as Decentralized$_i$.

| Strategy | Recall@1 (ACC) | | | | | | Recall@5 (ACC) | | | | | |
|---|---|---|---|---|---|---|---|---|---|---|---|---|
| | C1 | C2 | C3 | C4 | C5 | Avg. | C1 | C2 | C3 | C4 | C5 | Avg. |
| Centralized | 43.6 | 38.6 | 40.1 | 43.1 | 41.9 | 44.4 | 86.6 | 80.0 | 82.6 | 85.0 | 86.8 | 84.2 |
| FedAvg | 30.4 | 25.3 | 26.9 | 28.8 | 32.7 | 28.8 | 76.4 | 66.7 | 69.8 | 73.8 | 73.9 | 72.1 |
| Decentralized$_1$ | 17.7 | 14.7 | 15.4 | 18.2 | 14.1 | 16.0 | 57.0 | 49.6 | 51.3 | 54.6 | 55.1 | 53.5 |
| Decentralized$_2$ | 15.3 | 11.6 | 13.9 | 15.0 | 13.8 | 13.9 | 54.8 | 41.9 | 46.2 | 47.9 | 45.6 | 47.3 |
| Decentralized$_3$ | 17.4 | 13.1 | 14.1 | 15.1 | 15.4 | 15.0 | 50.4 | 44.2 | 46.4 | 49.7 | 52.5 | 48.6 |
| Decentralized$_4$ | 16.5 | 14.3 | 14.1 | 15.4 | 17.4 | 15.5 | 57.0 | 45.7 | 47.5 | 52.4 | 57.6 | 52.0 |
| Decentralized$_5$ | 21.7 | 14.3 | 14.2 | 15.6 | 19.5 | 17.1 | 57.9 | 48.4 | 50.2 | 52.4 | 61.7 | 54.1 |
| Local.avg. | 17.7 | 13.6 | 14.3 | 15.8 | 16.0 | 15.5 | 55.4 | 46.0 | 48.3 | 51.4 | 51.1 | 51.1 |

Table 14: The performance of the server model after 25 commu. turns and the averaged performance of corresponding local models after 25 and 26 commu. turns, on each client denoted as $\{C_i\}_{i=1}^5$. We utilize the ConVIRT as the backbone. We report the models after local update on client datasets of FedAvg pre-training strategy as $Local_i$.

| Strategy | com. turn | Recall@1 (ACC) | | | | | Recall@5 (ACC) | | | | |
|---|---|---|---|---|---|---|---|---|---|---|---|
| | | C1 | C2 | C3 | C4 | C5 | C1 | C2 | C3 | C4 | C5 |
| FedAvg | 25 | 30.4 | 25.3 | 26.9 | 28.8 | 32.7 | 76.4 | 66.7 | 69.8 | 73.8 | 73.9 |
| $Local_0$ | 25 | 31.8 | 23.4 | 26.0 | 26.2 | 33.7 | 73.4 | 64.1 | 67.2 | 71.4 | 71.8 |
| $Local_1$ | 25 | 27.7 | 22.5 | 23.8 | 25.1 | 27.7 | 73.4 | 63.1 | 64.6 | 69.2 | 68.6 |
| $Local_2$ | 25 | 28.6 | 24.4 | 24.3 | 27.3 | 28.9 | 73.3 | 64.9 | 67.5 | 70.3 | 71.8 |
| $Local_3$ | 25 | 30.6 | 22.6 | 23.9 | 25.4 | 26.4 | 72.6 | 64.0 | 66.3 | 68.9 | 67.6 |
| $Local_4$ | 25 | 27.3 | 24.4 | 25.5 | 26.5 | 28.9 | 73.3 | 65.8 | 69.0 | 71.2 | 69.2 |
| 1-5 Avg. | 25 | 29.2↓ | 23.5↓ | 24.7↓ | 26.1↓ | 29.1↓ | 73.2↓ | 64.4↓ | 66.9↓ | 70.2↓ | 69.8↓ |
| $Local_0$ | 26 | 29.9 | 23.2 | 25.2 | 26.7 | 33.1 | 73.4 | 64.1 | 67.2 | 71.4 | 71.8 |
| $Local_1$ | 26 | 28.7 | 23.1 | 24.0 | 25.9 | 27.4 | 74.5 | 63.4 | 64.1 | 69.7 | 67.9 |
| $Local_2$ | 26 | 30.9 | 23.9 | 25.4 | 27.7 | 29.9 | 72.4 | 65.6 | 67.4 | 71.6 | 71.1 |
| $Local_3$ | 26 | 30.1 | 22.7 | 23.4 | 24.5 | 27.1 | 73.1 | 63.5 | 65.3 | 67.5 | 67.3 |
| $Local_4$ | 26 | 27.0 | 24.8 | 25.5 | 26.6 | 29.6 | 73.6 | 66.0 | 69.2 | 71.2 | 69.5 |
| 1-5 Avg. | 26 | 29.3↓ | 23.5↓ | 24.7↓ | 26.3↓ | 29.4↓ | 73.3↓ | 64.4↓ | 66.3↓ | 70.1↓ | 69.4↓ |

Table 15: The accuracy of the test set of each client. We show the performance of FedAvg pre-trained baseline and its retrained models on different client datasets. We report the models after local update on client datasets of FedAvg pre-training strategy as $Local_i$.

| position | model | com. | Recall@1 (ACC) | | | | | | Recall@5 (ACC) | | | | | |
|---|---|---|---|---|---|---|---|---|---|---|---|---|---|---|
| | | | C0 | C1 | C2 | C3 | C4 | Avg. | C0 | C1 | C2 | C3 | C4 | Avg. |
| - | server | 25 | 30.4 | 25.3 | 26.9 | 28.8 | 32.7 | 28.8 | 76.4 | 66.7 | 69.8 | 73.8 | 73.9 | 72.1 |
| - | server | 50 | 32.3 | 26.0 | 27.0 | 27.1 | 30.2 | 28.5↓ | 77.6 | 67.9 | 69.4 | 72.1 | 71.7 | 71.7↓ |
| → shallow | $Local_0$ | 25 | 30.4 | 25.0 | 25.3 | 28.4 | 28.6 | 27.5↓ | 73.8 | 67.2 | 68.4 | 72.0 | 73.0 | 70.9↓ |
| → shallow | $Local_1$ | 25 | 34.3 | 26.4 | 27.3 | 29.7 | 30.2 | 29.4↑ | 78.3 | 69.8 | 72.3 | 75.1 | 77.4 | 74.6↑ |
| → shallow | $Local_2$ | 25 | 33.7 | 26.4 | 27.3 | 29.7 | 30.2 | 29.4↑ | 77.3 | 67.4 | 70.7 | 74.2 | 70.5 | 72.0↑ |
| → shallow | $Local_3$ | 25 | 27.7 | 18.9 | 19.3 | 25.6 | 24.9 | 25.0↓ | 72.4 | 64.2 | 64.8 | 70.5 | 71.1 | 68.6↓ |
| → shallow | $Local_4$ | 25 | 26.2 | 18.9 | 19.3 | 22.7 | 22.7 | 22.0↓ | 69.3 | 56.8 | 58.9 | 63.3 | 64.5 | 62.6↓ |

Table 16: The accuracy of the model on each client. We show the acc. of centralized and FedAvg pre-trained baselines and de-centralized pre-trained models shown as $Local_i$ retrained on the union of training splits of client datasets. We fine-tune shallow layers of the de-centralized pre-trained model with the union dataset.

| strategy | model | Recall@1 (ACC) | | | | | | Recall@5 (ACC) | | | | | |
|---|---|---|---|---|---|---|---|---|---|---|---|---|---|
| | | C1 | C2 | C3 | C4 | C5 | Avg. | C1 | C2 | C3 | C4 | C5 | Avg. |
| Global | server | 43.6 | 38.6 | 40.1 | 43.1 | 41.9 | 41.5 | 86.6 | 80.0 | 82.6 | 85.0 | 86.8 | 84.2 |
| FedAvg | server | 30.4 | 25.3 | 26.9 | 28.8 | 32.7 | 28.8 | 76.4 | 66.7 | 69.8 | 73.8 | 73.9 | 72.1 |
| Decentralized | $Local_1$ | 28.7 | 22.6 | 23.5 | 22.3 | 24.6 | 24.4 | 70.9 | 60.8 | 63.3 | 62.4 | 63.4 | 64.2 |
| Decentralized | $Local_2$ | 17.4 | 19.9 | 18.2 | 17.6 | 17.7 | 18.1 | 52.7 | 56.1 | 55.1 | 54.0 | 53.5 | 54.3 |
| Decentralized | $Local_3$ | 20.9 | 20.7 | 26.0 | 21.0 | 22.3 | 22.2 | 58.9 | 58.1 | 65.3 | 58.2 | 59.0 | 59.9 |
| Decentralized | $Local_4$ | 20.9 | 20.1 | 20.7 | 25.6 | 21.1 | 21.7 | 59.1 | 56.9 | 57.8 | 64.7 | 58.7 | 59.4 |
| Decentralized | $Local_5$ | 21.8 | 19.5 | 22.0 | 20.9 | 31.5 | 23.2 | 60.7 | 57.0 | 59.8 | 60.2 | 74.1 | 62.4 |
| Decentralized | $Local_1$ | 17.7 | 14.7 | 15.4 | 18.2 | 14.1 | 16.0 | 57.0 | 49.6 | 51.3 | 54.6 | 55.1 | 53.5 |
| Decentralized | $Local_2$ | 15.3 | 11.6 | 13.9 | 15.0 | 13.8 | 13.9 | 54.8 | 41.9 | 46.2 | 47.9 | 45.6 | 47.3 |
| Decentralized | $Local_3$ | 17.4 | 13.1 | 14.1 | 15.1 | 15.4 | 15.0 | 50.4 | 44.2 | 46.4 | 49.7 | 52.5 | 48.6 |
| Decentralized | $Local_4$ | 16.5 | 14.3 | 14.1 | 15.4 | 17.4 | 15.5 | 57.0 | 45.7 | 47.5 | 52.4 | 57.6 | 52.0 |
| Decentralized | $Local_5$ | 21.7 | 14.3 | 14.2 | 15.6 | 19.5 | 17.1 | 57.9 | 48.4 | 50.2 | 52.4 | 61.7 | 54.1 |

# D    Theoretical Analysis

## D.1    Derivation of Proposition 1

To begin, we will establish the following lemma.

**Lemma 1.** *Let $\{(z_{a,i}, z_{b,i})\}_{i=1}^{\text{bz}}$ be a mini-batch of* unit-norm *embeddings ($\|z_{a,i}\|_2 = \|z_{b,i}\|_2 = 1$) with batch size* bz *and temperature $\tau > 0$. Denote the average squared Euclidean distance*

$$\overline{D}^2 = \frac{1}{\text{bz}} \sum_{i=1}^{\text{bz}} \|z_{a,i} - z_{b,i}\|_2^2.$$

*Then the InfoNCE (contrastive) loss*

$$L_{\text{CL}}(z_a, z_b) = \frac{1}{\text{bz}} \sum_{i=1}^{\text{bz}} \left[ -\log \frac{\exp\big(\text{sim}(z_{a,i}, z_{b,i})/\tau\big)}{\sum_{j=1}^{\text{bz}} \exp\big(\text{sim}(z_{a,i}, z_{b,j})/\tau\big)} \right],$$

*with $\text{sim}(u, v) = u^\top v$, satisfies*

$$L_{\text{CL}}(z_a, z_b) \leq \log(\text{bz}) + \frac{\overline{D}^2}{2\tau}.$$

*Proof.* For unit vectors, $\|z_{a,i} - z_{b,i}\|_2^2 = 2 - 2\,\text{sim}(z_{a,i}, z_{b,i})$, so $\text{sim}(z_{a,i}, z_{b,i}) = 1 - \frac{1}{2}d_i^2$, where $d_i^2 := \|z_{a,i} - z_{b,i}\|_2^2 \in [0, 2]$.

Fix $i$. Because all cosine similarities are at most 1,

$$\sum_{j=1}^{\text{bz}} \exp\big(\text{sim}(z_{a,i}, z_{b,j})/\tau\big) \leq \text{bz}\, e^{1/\tau}.$$

Hence the $i$-th term of the loss obeys

$$-\log \frac{\exp\big((1 - \frac{1}{2}d_i^2)/\tau\big)}{\sum_j \exp(\text{sim}_{ij}/\tau)} \leq \log(\text{bz}) + \frac{d_i^2}{2\tau}.$$

Averaging over the batch and noting $\frac{1}{\text{bz}} \sum_i d_i^2 = \overline{D}^2$ gives the desired bound. $\qquad\square$

Using this lemma, we will complete the proof of Proposition 1. In this paper, without loss of generalizability, we assume $\mathcal{R}_T(\cdot)$ to be the contrastive loss.

*Proof.* We begin by expressing the generalization error $\mathcal{R}_T(\hat{f})$ on the target domain $\mathcal{D}_\mathcal{T}$ as the expected contrastive loss:

$$\mathcal{R}_T(\hat{f}) = \mathbb{E}_{(x,y) \sim \mathcal{D}_\mathcal{T}} \left[ L_{\text{CL}}(\hat{f}_\psi(x), \hat{f}_\phi(y)) \right], \tag{3}$$

where $L_{\text{CL}}$ is the contrastive loss defined as:

$$L_{\text{CL}}(z_a, z_b) = \frac{1}{\text{bz}} \sum_{i=1}^{\text{bz}} \left[ -\log \frac{\exp\left(\text{sim}(z_{a,i}, z_{b,i})/\tau\right)}{\sum_{j=1}^{\text{bz}} \exp\left(\text{sim}(z_{a,i}, z_{b,j})/\tau\right)} \right], \tag{4}$$

with $\text{sim}(z_{a,i}, z_{b,i}) = \frac{z_{a,i}^\top z_{b,i}}{\|z_{a,i}\| \|z_{b,i}\|}$ and bz being the batch size.

By Lemma 1, we have an upper bound on the contrastive loss:

$$L_{\text{CL}}(z_a, z_b) \leq \log(\text{bz}) + \alpha \overline{D}^2, \tag{5}$$

where $\overline{D}^2$ is the average squared Euclidean distance between $z_{a,i}$ and $z_{b,i}$:

$$\overline{D}^2 = \frac{1}{\text{bz}} \sum_{i=1}^{\text{bz}} \|z_{a,i} - z_{b,i}\|_2^2, \tag{6}$$

and $\alpha = \frac{1}{2\tau}$.

Applying this to our generalization error:

$$\mathcal{R}_T(\hat{f}) \leq \log(\text{bz}) + \alpha \mathbb{E}_{(x,y)\sim\mathcal{D}_\mathcal{T}}\left[\|\hat{f}_\psi(x) - \hat{f}_\phi(y)\|_2^2\right]. \tag{7}$$

Then, we have:

$$
\begin{aligned}
|\hat{f}_\psi(x) - \hat{f}_\phi(y)|_2^2 &\leq \Big(|\hat{f}_\psi(x) - \hat{f}_{\psi_i}(x)|_2 + |\hat{f}_{\psi_i}(x) - f_{\psi_i}(x)|_2 \\
&\quad + |f_{\psi_i}(x) - f_{\phi_i}(y)|_2 + |f_{\phi_i}(y) - \hat{f}_{\phi_i}(y)|_2 + |\hat{f}_{\phi_i}(y) - \hat{f}_\phi(y)|_2\Big)^2 \\
&\leq 5\Big(|\hat{f}_\psi(x) - \hat{f}_{\psi_i}(x)|_2^2 + |\hat{f}_{\psi_i}(x) - f_{\psi_i}(x)|_2^2 \\
&\quad + |f_{\psi_i}(x) - f_{\phi_i}(y)|_2^2 + |f_{\phi_i}(y) - \hat{f}_{\phi_i}(y)|_2^2 + |\hat{f}_{\phi_i}(y) - \hat{f}_\phi(y)|_2^2\Big),
\end{aligned}
\tag{8}
$$

where the last inequality follows from the fact that for any real numbers $a_1, \ldots, a_n$,

$$\left(\sum_{j=1}^n a_j\right)^2 \leq n \sum_{j=1}^n a_j^2.$$

Define the error terms:

$$
\begin{aligned}
\epsilon_{\hat{f}_\psi, \hat{f}_{\psi_i}}(x) &= |\hat{f}_\psi(x) - \hat{f}_{\psi_i}(x)|_2^2, \\
\epsilon_{\hat{f}_{\psi_i}, f_{\psi_i}}(x) &= |\hat{f}_{\psi_i}(x) - f_{\psi_i}(x)|_2^2, \\
\epsilon_{\hat{f}_\phi, \hat{f}_{\phi_i}}(y) &= |\hat{f}_\phi(y) - \hat{f}_{\phi_i}(y)|_2^2, \\
\epsilon_{\hat{f}_{\phi_i}, f_{\phi_i}}(y) &= |\hat{f}_{\phi_i}(y) - f_{\phi_i}(y)|_2^2, \\
C_i(x,y) &= |f_{\psi_i}(x) - f_{\phi_i}(y)|_2^2.
\end{aligned}
$$

Then inequality (8) becomes:

$$|\hat{f}_\psi(x) - \hat{f}_\phi(y)|_2^2 \leq 5\left(\epsilon_{\hat{f}_\psi, \hat{f}_{\psi_i}}(x) + \epsilon_{\hat{f}_{\psi_i}, f_{\psi_i}}(x) + C_i(x,y) + \epsilon_{\hat{f}_{\phi_i}, f_{\phi_i}}(y) + \epsilon_{\hat{f}_\phi, \hat{f}_{\phi_i}}(y)\right). \tag{9}$$

Taking expectation over $(x,y) \sim \mathcal{D}_\mathcal{T}$ and using $\mathcal{D}_\mathcal{T} = \sum_{i=1}^N w_i \mathcal{D}_i$, we have:

$$
\begin{aligned}
\mathbb{E}_{(x,y)\sim\mathcal{D}_\mathcal{T}}\left[|\hat{f}_\psi(x) - \hat{f}_\phi(y)|_2^2\right] &= \sum_{i=1}^N w_i\, \mathbb{E}_{(x,y)\sim\mathcal{D}_i}\left[|\hat{f}_\psi(x) - \hat{f}_\phi(y)|_2^2\right] \\
&\leq 5\sum_{i=1}^N w_i\, \mathbb{E}_{(x,y)\sim\mathcal{D}_i}\Big[\epsilon_{\hat{f}_\psi, \hat{f}_{\psi_i}}(x) + \epsilon_{\hat{f}_{\psi_i}, f_{\psi_i}}(x) \\
&\quad + C_i(x,y) + \epsilon_{\hat{f}_{\phi_i}, f_{\phi_i}}(y) + \epsilon_{\hat{f}_\phi, \hat{f}_{\phi_i}}(y)\Big].
\end{aligned}
\tag{10}
$$

Define $C_i = \mathbb{E}_{(x,y)\sim\mathcal{D}_i}\left[C_i(x,y)\right]$. Then,

$$
\begin{aligned}
\mathbb{E}_{(x,y)\sim\mathcal{D}_\mathcal{T}}\left[|\hat{f}_\psi(x) - \hat{f}_\phi(y)|_2^2\right] &\leq 5\sum_{i=1}^N w_i\Big(\mathbb{E}_{x\sim\mathcal{D}_i}\left[\epsilon_{\hat{f}_\psi, \hat{f}_{\psi_i}}(x) + \epsilon_{\hat{f}_{\psi_i}, f_{\psi_i}}(x)\right] \\
&\quad + \mathbb{E}_{y\sim\mathcal{D}_i}\left[\epsilon_{\hat{f}_{\phi_i}, f_{\phi_i}}(y) + \epsilon_{\hat{f}_\phi, \hat{f}_{\phi_i}}(y)\right] + C_i\Big).
\end{aligned}
\tag{11}
$$

Substituting (11) into the generalization error bound, we obtain:

$$\mathcal{R}_T(\hat{f}) \leq \log(\text{bz}) + \alpha \sum_{i=1}^{N} w_i \left( 5\mathbb{E}_{x \sim \mathcal{D}_i} \left[ \epsilon_{\hat{f}_\psi, \hat{f}_{\psi_i}}(x) + \epsilon_{\hat{f}_{\psi_i}, f_{\psi_i}}(x) \right] \right.$$
$$\left. + 5\mathbb{E}_{y \sim \mathcal{D}_i} \left[ \epsilon_{\hat{f}_{\phi_i}, f_{\phi_i}}(y) + \epsilon_{\hat{f}_\phi, \hat{f}_{\phi_i}}(y) \right] + 5C_i \right). \tag{12}$$

Letting $\alpha_i = 5\alpha$, we have:

$$\mathcal{R}_T(\hat{f}) \leq \sum_{i=1}^{N} w_i \alpha_i \left( \mathbb{E}_{x \sim \mathcal{D}_i} \left[ \epsilon_{\hat{f}_\psi, \hat{f}_{\psi_i}}(x) + \epsilon_{\hat{f}_{\psi_i}, f_{\psi_i}}(x) \right] + \mathbb{E}_{y \sim \mathcal{D}_i} \left[ \epsilon_{\hat{f}_{\phi_i}, f_{\phi_i}}(y) + \epsilon_{\hat{f}_\phi, \hat{f}_{\phi_i}}(y) \right] + C_i \right) + \log(\text{bz}). \tag{13}$$

Since $\log(\text{bz})$ is independent of $\hat{f}$, it can be considered a constant. Thus, we can express the generalization error as:

$$\mathcal{R}_T(\hat{f}) \leq \sum_{i=1}^{N} w_i \alpha_i \left( \epsilon_{\hat{f}_\psi, \hat{f}_{\psi_i}} + \epsilon_{\hat{f}_{\psi_i}, f_{\psi_i}} + \epsilon_{\hat{f}_\phi, \hat{f}_{\phi_i}} + \epsilon_{\hat{f}_{\phi_i}, f_{\phi_i}} + C_i \right).$$

This completes the proof of Proposition 1. □