# OpenReview forum: "Distributionally Robust Alignment for Medical Federated Vision-Language Pre-training Under Data Heterogeneity"
_TMLR — Accepted by TMLR_

### Review · Reviewer_QCdX · 2024-12-15

**Summary Of Contributions:**

This paper proposes a novel federated learning method for vision and language pretraining aimed for the medical application. To make the learned model robust to the unknown test domain, the author proposes to (i) separate the model into encoder and aligner, (ii) constrain the local weight using the global model and (iii) train them in the two-stage training setting and train the model with robust loss. The author experimentally demonstrates the effectiveness of the proposed method on the medical multi-modal data.

**Audience:**

Yes

**Broader Impact Concerns:**

Federated learning is aimed for the privacy-preserving model training. Emphasizing that the information communicated between server and clients does not harm to the client privacy would help user adopt the method with safety.

**Claims And Evidence:**

Yes

**Requested Changes:**

Modification of the Algorithms would help increase the clarity of the paper.

Adding the score deviation would be important to verify the significance of the score increase.

**Strengths And Weaknesses:**

Strength

The author proposes a novel federated learning method for training vision and language pretraining model in the server client setting.

The author experimentally demonstrates the ineffectiveness of the existing average model under heterogeneity. The proposed method shows better accuracy than the methods in the most settings.

The author conducts ablation study to show the effectiveness of the proposed modules. Further hyperparameter analysis is conducted.


Weakness

The paper says that DRO loss is used In both the first stage and second stage, but it looks like the loss is different between Algorithm 2 and Algorithm 3. I think it would be better to match the losses in the Algorithms if they are the same. Further, I think it would be better to include the DRO process in the Algorithm since it is a little complex and involves server-client communication.

In some case, the performance change is the order of 0.1% accuracy. I would like to see the standard deviation of the scores to verify the significance of the result.

---

### Review · Reviewer_ecgm · 2025-01-13

**Summary Of Contributions:**

This work addresses data-heterogeneity challenge in federated learning for medical multi-modal learning task. It proposes a two-stage global-guided local training strategy along-with Distributionally-Robust-Optimization (DRO) weighting to learn robust and enriched
cross-modal alignment, and better generalization.

**Audience:**

Yes

**Broader Impact Concerns:**

No ethical concern.

**Claims And Evidence:**

Yes

**Requested Changes:**

The performance gain across all the tasks (even on multi-modal task) with the other baseline methods (table 2) seem really small. Have the authors done multiple runs to report the standard error, it is important to see if the performance improvement is significant as compared to other methods such as FedAvg.

Even in ablation studies table, it is difficult to observe the gains,  for example absence of two-stage component consistently shows less than 1 % performance drop.

Authors should also perform a computational overhead (training time) comparison with the other baselines.

**Strengths And Weaknesses:**

The experiments are extensive. It covers classification, segmentation and image retrieval task. The proposed design choices seem reasonable and are backed by proper ablation studies.

---

### Review · Reviewer_osd7 · 2025-01-20

**Summary Of Contributions:**

The paper proposes FedDRA (Federated Distributionally Robust Alignment), a VLP (Vision-Language Pre-training) framework for medical applications that uses federated learning to scale up datasets while preserving privacy. FedDRA addresses data heterogeneity challenges in real-world scenarios by constructing a distribution family encompassing potential test-time domains and using a distributionally robust framework. The framework includes a two-stage approach to avoid overfitting on client-specific information, first tuning deeper layers before updating the entire network. The server optimizes the final model towards the target set distribution, with mirror gradient descent to set the according weights $\lamba_i$ for training the local modules.

**Audience:**

Yes

**Claims And Evidence:**

Yes

**Requested Changes:**

Can the authors specifically address the following?
- Address the computational cost that may be significant compared to the other baselines such as FedAvg
- Can the authors comment in the paper on the generalizability of this method to other domains?
- Can the authors also address on cases when the target set distribution is unknown (which is often the case in FL) and how effective the proposed method may be in this case?
- It would be nice if the paper was reorganized, especially the proposed method part since the method is quite complex, and reading through the paper was quite difficult with many things to parse.

**Strengths And Weaknesses:**

**Strengths**
- The paper addresses a significant and timely problem in medical VLP: data heterogeneity in federated learning settings.  This is a relatively unexplored area with high practical relevance for real-world medical applications.
- The proposed FedDRA method appears technically sound, employing a distributionally robust optimization framework and a two-stage training approach to address the challenges of data heterogeneity and overfitting.
- The paper provides extensive experiments on real-world medical datasets, demonstrating the effectiveness of FedDRA in enhancing medical federated VLP under data heterogeneity.  The results show consistent improvements over baseline methods.

**Weaknesses**
- Clarity and Presentation: The paper could benefit from improved clarity and presentation. Some technical details are not clearly explained, and the figures could be more informative. More specific examples and clearer illustrations would enhance the paper's readability and impact.
- Limited Scope: The paper focuses on a specific type of VLP (image-text contrastive learning) and a limited set of medical applications. The generalizability of the proposed method to other VLP approaches and medical domains is not fully explored.
- Computational Cost: The paper does not provide a detailed analysis of the computational cost of FedDRA. The two-stage training approach and the distributionally robust optimization framework could potentially increase the computational burden, especially for resource-constrained clients.

---

### Decision · Action_Editor_1Fw5 · 2025-05-01

**Recommendation:** Accept with minor revision

**Comment:**

This submission proposes Federated Distributionally Robust Alignment (FedDRA), a Vision-Language Pre-training framework for medical applications that uses federated learning to scale up datasets while preserving privacy. The proposed method addresses data heterogeneity challenges in real-world scenarios by constructing a distribution family encompassing potential test-time domains and using a distributionally robust framework. The framework includes a two-stage approach to avoid overfitting on client-specific information.

In consideration of the above contributions, all the reviewers are basically positive to this work, which are strengthened by the authors' replies on concerns about clarity, presentation, computational cost, DRO loss, and marginal accuracy gain. The paper can be further improved by minor revisions based on the reviewers' comments.

**Audience:**

Yes. Researchers in the federated learning for medical applications should be interested in this work, in order to train good models in case of heterogeneious real-world data.

**Claims And Evidence:**

The reviewers recognized some strengths in the paper including: 1) The problem addressed, i.e. data heterogeneity in federated learning, is significant and worth studying, as a relatively unexplored area good for real-world medical applications. 2) The proposed method is technically sound, which employs a distributionally robust optimization framework and a two-stage training approach to address the challenges of data heterogeneity and overfitting.

The claims are validated by extensive experiments on real-world medical datasets, demonstrating the effectiveness of the proposed method in enhancing medical federated VLP under data heterogeneity.